# CNV-association meta-analysis in 191,161 European adults reveals new loci associated with anthropometric traits

Aurélien Macé *et al.*[#]

There are few examples of robust associations between rare copy number variants (CNVs) and complex continuous human traits. Here we present a large-scale CNV association meta-analysis on anthropometric traits in up to 191,161 adult samples from 26 cohorts. The study reveals five CNV associations at 1q21.1, 3q29, 7q11.23, 11p14.2, and 18q21.32 and confirms two known loci at 16p11.2 and 22q11.21, implicating at least one anthropometric trait. The discovered CNVs are recurrent and rare (0.01–0.2%), with large effects on height (>2.4 cm), weight (>5 kg), and body mass index (BMI) (>3.5 kg/m$^2$). Burden analysis shows a 0.41 cm decrease in height, a 0.003 increase in waist-to-hip ratio and increase in BMI by 0.14 kg/m$^2$ for each Mb of total deletion burden ($P = 2.5 \times 10^{-10}$, $6.0 \times 10^{-5}$, and $2.9 \times 10^{-3}$). Our study provides evidence that the same genes (e.g., *MC4R*, *FIBIN*, and *FMO5*) harbor both common and rare variants affecting body size and that anthropometric traits share genetic loci with developmental and psychiatric disorders.

#A full list of authors and their affiliations appears at the end of the paper.

Many human anthropometric traits are highly heritable. Twin studies have estimated that genetic factors contribute to 40–80% of the observed variability of body mass index (BMI)[1–5] and up to 80% of height[6, 7]. Findings from the largest genome-wide association studies (GWAS) on BMI[8] and height[9], including over 250,000 samples, revealed 97 and 697 single nucleotide polymorphisms (SNPs) explaining cumulatively only 2.7 and 20% of the variance of the respective phenotypes. Using genotyping arrays enriched for coding regions (exome-chip) large meta-analysis GWAS for height and BMI discovered several rare coding single nucleotide variants (SNVs) associated with these traits. Still, these SNVs have thus far explained only a very small variation in these traits (e.g., 0.51% explained height variance[10]). Nevertheless, random effect models accounting for imperfect imputation estimate that the total additive effect of all SNVs explain 56 and 27% of height and BMI variability, respectively[11]. While there is a growing consensus that predominantly SNVs contribute to the heritability, the impact of the structural architecture of the genome (copy number variants, complex rearrangements, etc.) is understudied and not negligible[12]. It has been shown that rare and large copy number variants (CNVs), such as the 600 kb breakpoint 4–5 (BP4–BP5) 16p11.2 rearrangement[13, 14], can exert substantial impact on BMI, but little effort has been made towards assessing the genome-wide impact of CNVs on complex traits. To our knowledge, only one genome-wide CNV-association study (on schizophrenia) has been performed in large adult population samples[15]. The aim of our study is to establish a genome-wide catalog of CNVs and to identify CNVs associated with height, weight, waist-to-hip ratio (WHR) and BMI. To this end, we apply the same CNV calling[16] and association pipeline to 25 studies of the Genetic Investigation of Anthropometric Traits (GIANT) Consortium combined with the UK Biobank and perform a genome-wide association meta-analysis study in up to 191,161 unrelated European adults. These analyses show that overall CNV burden is linked to shorter stature and higher WHR. The genome-wide scans reveal rare variations in several genomic regions (1q21.1, 7q11.23, 3q29, 16p11.2, *FIBIN/BBOX1*, and *MC4R*) to be associated with anthropometric measures. Some of these loci have variable frequencies across cohorts and explain or overlap previous SNP or rare variant associations. These results highlight the important contribution of rare CNVs to complex human traits.

## Results

**Summary of the methods**. All the 25 GIANT cohorts were genotyped on Illumina arrays, whereas the UK Biobank used the Affymetrix Axiom chip. Only unrelated adult samples of European origin were included. As PennCNV was initially designed for data generated on Illumina arrays, we took extra care with the signal normalization and pre-processing of the UK Biobank data (see "Methods"). Each cohort applied our standardised CNV pipeline to call CNVs[16] and to test associations between probabilistic CNV dosages (a continuous value between −1 (deletion) and 1 (duplication)) at each probe on the genotyping chip and our target anthropometric traits. In brief, our pipeline combines pennCNV calls, CNV- and sample parameters to yield a more accurate probabilistic CNV call, especially in the case of rare or low confidence CNVs. The number of probes varied between ~680,000 and 2,500,000 across the 26 cohorts. We then imputed the summary statistics to the Illumina 1M Duo V3 probe set in order to have a common set of probes for the meta-analysis. Based on an in-house cohort we conservatively estimated the number of effective tests[17] to be ~29,400, resulting in a P-value threshold of $1.7 \times 10^{-6}$ to control family-wise error rate (see "Methods"). We performed a CNV burden and a genome-wide CNV association meta-analysis for BMI, weight, height, and waist–hip ratio. The genome-wide CNV association scan was performed considering a mirror effect model (assuming opposite and equal sized effect of deletions and duplications at any given locus) and the genome-wide significant signals were further tested for deletion-only and duplication-only effects. As secondary analysis, we also tested U-shaped (assuming the same effect of deletions and duplications), deletion-only, duplication-only models genome-wide. All reported CNV effect sizes (unless specified otherwise) represent the impact of one additional copy relative to the population average. For burden analysis all four abovementioned models (mirror, U-shaped, deletion, duplication) were tested. Depending on the trait, the sample sizes varied between 161,244 and 191,161.

**Total CNV burden**. The increased burden of rare CNVs has already been observed for persons with short stature[18], higher BMI[19], and also schizophrenia[15]. Indirectly, increased deletion burden is also reflected in longer regions with loss-of-heterozygosity, which has shown to associate with stature and

**Table 1 List of the CNVs associated with one or several traits**

| Chr | Start | End | Frequency (%) | | BMI | | Weight | | Height | | Waist–hip ratio | |
|---|---|---|---|---|---|---|---|---|---|---|---|---|
| | (Mb) | (Mb) | Del | Dup | β | P value | β | P value | β | P value | β | P value |
| 1 | 145 | 145.9 | 0.03 | 0.049 | – | – | 6.66 | 1.73E−06 | 3.46 | 3.75E−10 | – | – |
| 3 | 197.7 | 197.9 | 0.004 | 0.005 | – | – | 22.55 | 1.20E−06 | – | – | – | – |
| 3 | 198.2 | 198.4 | 0.007 | 0.007 | – | – | – | – | 13.3 | 2.32E−08 | – | – |
| 7 | 72.61 | 72.75 | 0.005 | 0.005 | – | – | – | – | – | – | 0.11 | 1.49E−06 |
| 11 | 26.97 | 27.19 | 0.126 | 0.011 | – | – | – | – | 2.43 | 1.46E−06 | – | – |
| 16 | 28.73 | 28.95 | 0.028 | 0.041 | −3.07 | 5.31E−08 | −10.35 | 5.03E−09 | – | – | – | – |
| 16 | 29.5 | 30.1 | 0.027 | 0.031 | −3.66 | 1.39E−12 | – | – | 5.21 | 1.20E−14 | −0.041 | 2.30E−07 |
| 18 | 55.81 | 56.05 | 0.018 | 0.004 | −5.06 | 2.03E−07 | 15.9 | 1.45E−08 | – | – | – | – |

All positions are hg18 in megabase (Mb). In case of genome-wide significant CNV-trait associations (P < 1.7 × 10⁻⁶), we report effect sizes (β) and P values coming from a mirror effect model, assuming opposite and equal sized effect of deletions and duplications at any given locus. Further information and results from other models are available in Supplementary Table 2. The effects correspond to change in the trait for each additional copy of the region: positive effect means that deletion of the corresponding region decreases the trait value and duplications increase it. The genes involved in these regions are as follows: RN7SL261P, RNVU1-8, CHD1L, NBPF13P, GJA8, OR13Z3P, LINC00624, OR13Z2P, OR13Z1P, PDIA3P1, FMO5, RPL7AP15, CCT8P1, PRKAB2, GJA5, GPR89B, BCL9, ACP6, (Chr1:145–146 Mb); PIGX, (Chr3:197.7–197.9 Mb); DLG1, MFI2, MFI2-AS1, (Chr3:198.2–198.4 Mb); VPS37D, DNAJC30, WBSCR22, MLXIPL, (Chr7:72.61–72.75 Mb); FIBIN, BBOX1, BBOX1-AS1, (Chr11:26.97–27.19 Mb); MIR4721, MIR4517, ATXN2L, SH2B1, CD19, RABEP2, TUFM, ATP2A1, NFATC2IP, ATP2A1-AS1, LAT, SPNS1, (Chr16:28.7–29 Mb); MIR3680-2, RN7SKP127, C16orf54, PAGR1, CORO1A, MAZ, ALDOA, CDIPT, MVP, ZG16, SEZ6L2, CDIPT-AS1, PRRT2, YPEL3, TMEM219, DOC2A, GDPD3, INO80E, KCTD13, HIRIP3, ASPHD1, MAPK3, TAOK2, PPP4C, FAM57B, C16orf92, SMG1P2, SLC7A5P1, CA5AP1, QPRT, SPN, TBX6, KIF22, (Chr16:29.5–30.1 Mb); RNU4-17P, RNU6-567P, SDCCAG3P1, FAM60CP, RPS3AP49, (Chr18:55.8–56.1 Mb).

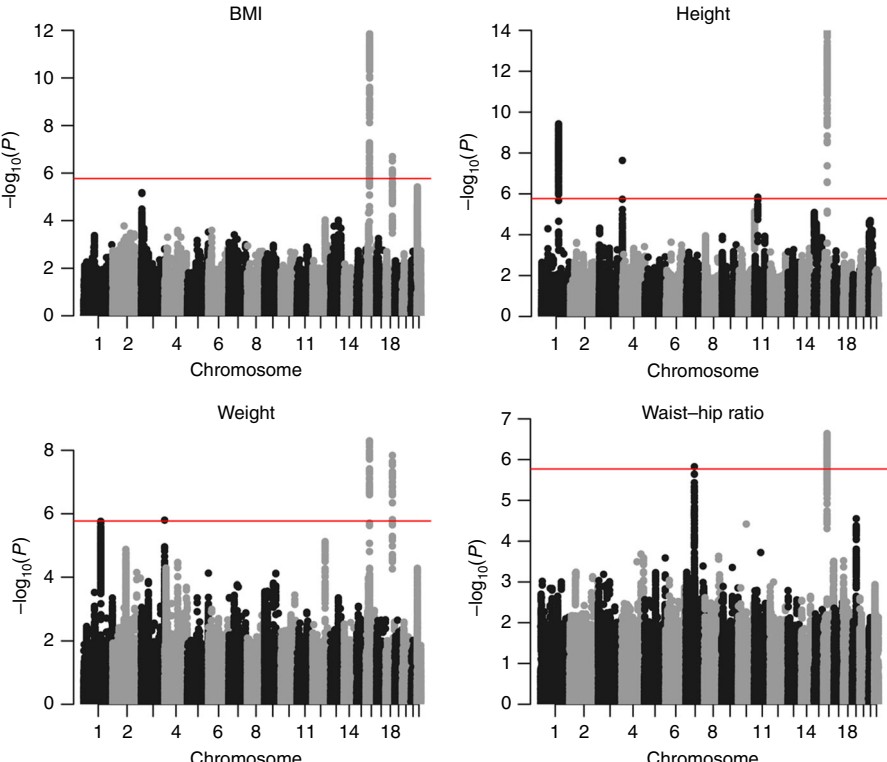

**Fig. 1** Genome-wide Manhattan plots for four anthropometric traits. Genome-wide association study for CNVs associated with BMI, height, weight, and waist-to-hip ratio in 191,161 Europeans

cognition[20]. In this study we confirm the link between CNV burden, measured as the total number of copy variant probes, and height and BMI and we also found an effect on the waist–hip ratio (Supplementary Table 1). Individuals with an additional 1 Mb of copy-altered interval tend to have 0.144 kg/m² higher BMI ($P = 2.9 \times 10^{-3}$). The effect of CNV burden was much stronger on waist–hip ratio and height, for which increased CNV burden (be it duplication or deletion) was associated with a 0.001 higher WHR ($P = 6.9 \times 10^{-5}$) and 0.132 cm shorter stature ($P = 4.5 \times 10^{-7}$). For both traits the impact was dominated by the burden of deletions rather than duplications (0.003 WHR unit ($P = 6 \times 10^{-5}$) and 0.41 cm ($P = 2.5 \times 10^{-10}$) per Mb deletion, respectively). We did not observe any CNV burden effect on human weight.

**Genome-wide scan**. The analyses on these four anthropometric traits revealed seven independent CNV regions associated with one or several traits with $P$ value below the genome-wide significance threshold ($1.7 \times 10^{-6}$, see "Methods") (Table 1, Fig. 1). Two of them correspond to the well known BP2–BP3 and BP4–BP5 CNVs in the 16p11.2 region associated with BMI and neurodevelopment. Three further CNVs (1q21.1, 3q29, and 7q11.23) overlap recurrent syndromic CNV regions, associated with variable neurodevelopmental traits, schizophrenia and developmental delay. One CNV (near *MC4R*) overlaps with SNPs associated with BMI in GWAS and is part of a larger deletion reported to be associated with obesity[21]. And finally one deletion, encompassing *BBOX1* and *FIBIN* genes (the latter harboring rare, height-lowering coding variants[10]), seems to be particularly frequent in the Finnish population (0.89% vs 0.02% in the non-Finnish cohorts). In the following we provide a detailed description of the impact of each of these CNVs (both deletions and duplications). We tested U-shaped, deletion-only, duplication-only models, but these did not yield further significant associations (Supplementary Table 2).

**New insights on the 16p11.2 region**. The 16p11.2 region is well known for several distinct recurrent CNVs, two of them associated with anthropometric traits. The 220 kb BP2–BP3 deletion was associated with severe early-onset obesity and developmental delay[22]. The 600 kb BP4–BP5 rearrangement was first known for its impact on autism but it has also been proven to have effects on BMI and head circumference[13, 14]. Both have been recently reported as associated with lower IQ and schizophrenia[15]. First, we replicated the known effects of the 220 kb deletion ($\beta = +3.07$ kg/m², $P = 5.3 \times 10^{-8}$) and the mirror effect of the 600 kb rearrangement (mirror effect: $\beta = -3.66$ kg/m², $P = 1.4 \times 10^{-12}$; deletion: $\beta = 6.15$ kg/m², $P = 4.5 \times 10^{-14}$; duplication: $\beta = -1.81$ kg/m², $P = 1.2 \times 10^{-2}$) on BMI. In addition, we found that while the 220 kb deletion increases BMI through increasing weight (by 10.35 kg, $P = 5 \times 10^{-9}$), the 600 kb deletion does so by both decreasing height (by 5.21 cm, $P = 1.1 \times 10^{-14}$) and increasing weight (6.57 kg, $P = 5.3 \times 10^{-5}$) (Supplementary Figs. 1–4). Furthermore, our analysis revealed that the 600 kb rearrangement also impacts waist–hip ratio ($\beta = -0.04$, $P = 2.3 \times 10^{-7}$) (Supplementary Figs. 3C–4C). Neither analyzing deletions and duplications separately, nor their absolute effect showed stronger signal for the 16p11.2 220 kb rearrangement than the mirror effect association. On the contrary, the observed effect from the 600 kb seems almost exclusively driven by the deletion, which demonstrated a stronger signal than the duplications or the pooled results (Supplementary Table 2). The list of the Online Mendelian Inheritance in Man (OMIM) diseases corresponding to the genes present in these two CNVs is available in Supplementary Table 3. The top associations between these CNVs and 27 tested traits in the UK Biobank are listed in Supplementary Tables 4–7.

We could not narrow down the BMI association signal to the previously proposed *SH2B1* (lowest $P = 7.7 \times 10^{-8}$) as it covers other genes, including *SPNS1* and *LAT* (lowest $P = 5.3 \times 10^{-8}$) (Fig. 2). Fine-mapping of the signal using variable breakpoints would be necessary, which are extremely rare due to the regional

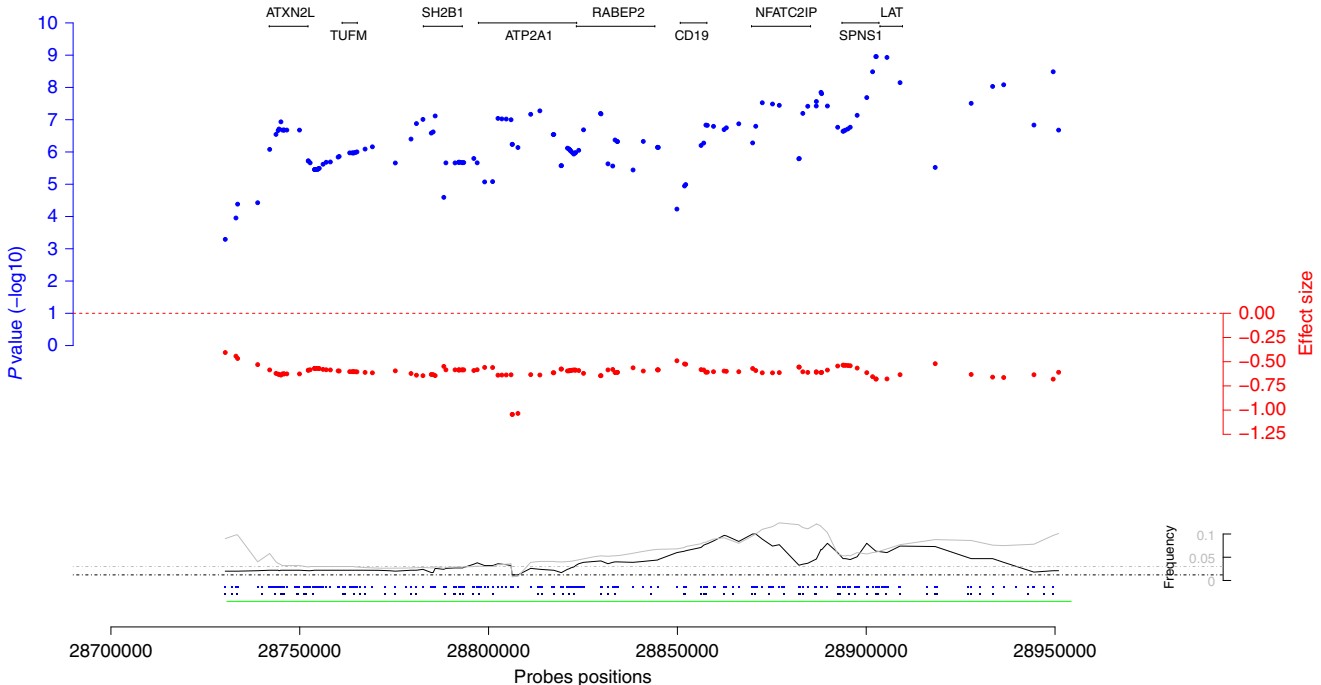

**Fig. 2** Regional association plot for the 16p11.2 220 kb rearrangement. The *blue dots* represent −log10 BMI-association *P* values, the *red dots* show the corresponding effect sizes. At the *bottom* the *black* and *gray lines* are the deletion and duplication frequencies. Finally, the *dots* at the *bottom* indicate the probe positions for the GIANT cohorts (*above*) and the UK Biobank (*below*). Positions of the protein-coding genes are shown at the *top* of the plot. The probes positions correspond to the human genome build 36

architecture shaped by segmental duplications and non-allelic homologous recombinations (Supplementary Fig. 2).

Results from previous GWAS revealed BMI-associated SNPs near *SH2B1* located in the 220 kb rearrangement and height-associated SNPs near *FLJ25404* located in the 600 kb rearrangement highlighting the importance of both common and rare variants in these regions (Supplementary Table 8). Next, we tested whether the previously published three independent BMI-associated SNPs at this locus (rs3888190, rs2650492, and rs4787491) could be explained by the CNV associations in the 16p11.2 region using the UK Biobank data for which both SNPs and CNV calls were available. Our analysis showed that the original BMI-SNP association *P* values increased substantially (from $2.28 \times 10^{-8}$, $6.45 \times 10^{-5}$, $8.67 \times 10^{-6}$ to $2.94 \times 10^{-4}$, $3.80 \times 10^{-3}$, $1.56 \times 10^{-2}$, respectively) when the most BMI-associated CNV probe was included in the multivariate model. In the meantime the BMI-CNV association signals remained unchanged. Similarly, the height-*FLJ25404* (rs11642612) association *P* value increased more than 100-fold from $P = 1.5 \times 10^{-5}$ to $P = 4.35 \times 10^{-3}$ when including the 16p11.2 CNV probe with strongest height association, indicating that the previously observed SNP-height association may be (at least partially) explained by the 16p11.2 CNV-height association.

*Cis* eQTL analysis for height- and BMI-associated SNPs located in the 600 kb rearrangement showed a potential effect of these SNPs modulating the expression levels (in whole-blood) of *CORO1A* (rs11150581, rs11642612) and *INO80E* (rs11150581, rs11642612, rs2278557, rs6565173, rs9925915) genes (Supplementary Table 9).

**1q21.1 distal rearrangement**. A CNV region on chromosome 1 (145–145.9 Mb, Supplementary Figs. 5 and 6) was associated with both height ($\beta = 3.46$ cm, $P = 3.8 \times 10^{-10}$) and weight ($\beta = 6.66$ kg, $P = 1.7 \times 10^{-6}$). This rearrangement corresponds to the distal part of the 1q21.1 recurrent CNV (OMIM deletion: #612474;

OMIM duplication: #612475). As for the 16p11.2 600 kb CNV, this CNV is known to have a mirror effect on head circumference and to be a potential cause of autism and schizophrenia[15, 23]. An effect on height has been reported for the deletion, with 25–50% of the carriers having short stature[24]. In contrast, duplication carriers tend to be in the upper percentiles of height but the effect is less clear. Supporting the effect on height, a common variant in this region, near the *FMO5* gene (rs6658763) was associated with height in previous GWAS[9] (Supplementary Table 8). This SNP was not significantly associated with height in the UK Biobank ($P = 0.14$), and thus, no conditional analysis was performed. However, the SNP seems to be independent of the CNV (Deletion: $r^2 = 0$, $D' = 0.005$–Duplication $r^2 = 0$, $D' = 0.022$ in the UK Biobank). As for the 16p11.2 600 kb rearrangement, the observed effects seem to be mainly due to the deletions (Supplementary Table 2). The list of the OMIM diseases corresponding to the genes present in the CNV is available in Supplementary Table 3. The top associations between this CNV and 27 tested traits in the UK Biobank are listed in Supplementary Tables 10 and 11.

**A CNV overlapping *FIBIN* and *BBOX1***. A 220 kb CNV (chr11: 26.97–27.19 Mb, Supplementary Figs. 7 and 8) was associated with height ($\beta = 2.43$ cm, $P = 1.5 \times 10^{-6}$). While the duplication frequency is low in all cohorts (0.008–0.016%), the deletion frequency is much higher in the Finnish population than in the others (0.89% vs 0.016%). This region has added interest, because a case-report described an Iranian short-statured girl with homozygous deletion of this region[25]. Separate analysis of the deletions and duplications showed a highly significant effect from the deletions ($\beta = 2.56$ cm, $P = 8.2 \times 10^{-8}$). The involvement of the *FIBIN* gene for height was also confirmed by the GIANT-exome study on height[10] including 381,625 individuals. This study revealed a strong association between the rare (0.3% in ExAC) missense variant rs138273386, located in the *FIBIN* gene, and height ($P = 5.79 \times 10^{-12}$) (Supplementary Table 12). The top

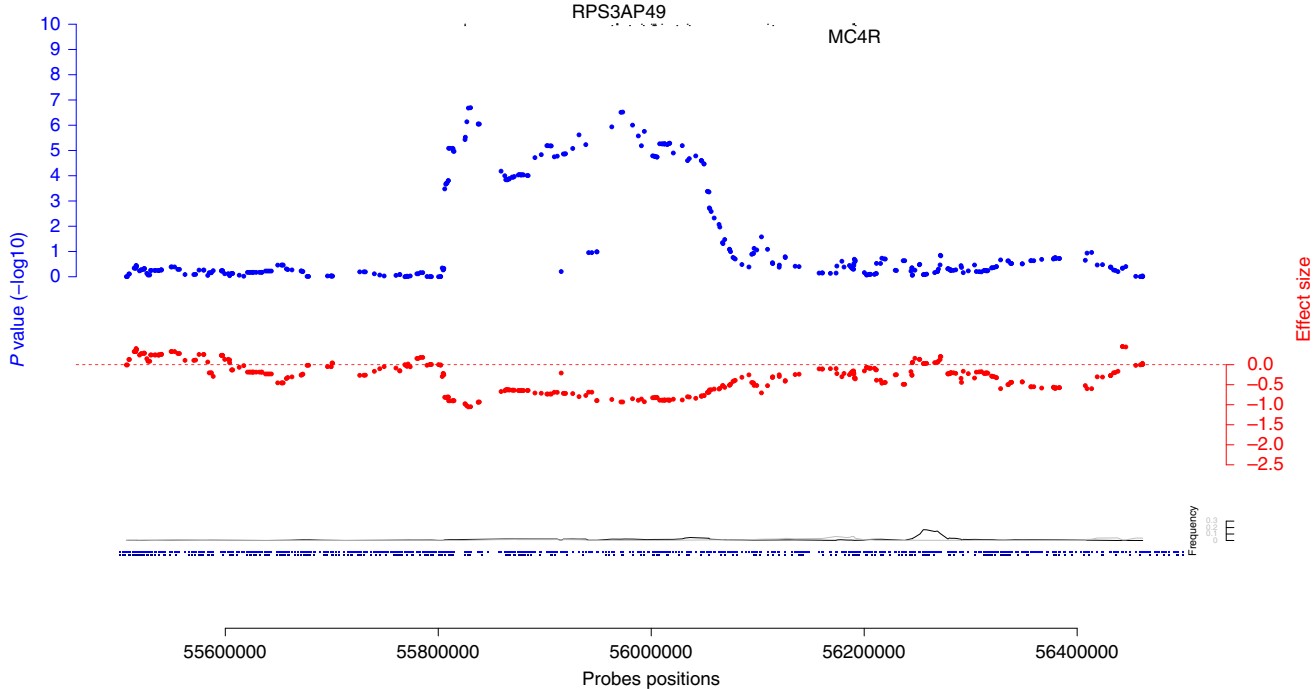

**Fig. 3** Regional association plot for the rearrangement near *MC4R*. The *blue dots* represent –log10 BMI-association *P* values, the *red dots* show the corresponding effect sizes. At the *bottom* the *black* and *gray lines* are the deletions and duplications frequencies. Finally, the *dots* at the *bottom* are the probes positions for the GIANT cohorts *above* and the UK BioBank *below*. Positions of the protein-coding genes are shown at the *top* of the plot along with the position of the BMI-associated GWAS SNP. The probes positions correspond to the human genome build 36

associations between this CNV and 27 tested traits in the UK Biobank are listed in Supplementary Tables 13 and 14.

**CNV in the *MC4R* region.** Single nucleotide coding mutations in the *MC4R* gene cause severe obesity, and common variants near the gene are associated with BMI[8, 26, 27]. Our analysis revealed a rare (frequency 0.018% (del), 0.004% (dup)), 300 kb long CNV (55.81–56.05 Mb, Supplementary Figs. 9 and 10) associated with BMI ($\beta = -5.06$ kg/m$^2$, $P = 2 \times 10^{-7}$) and weight ($\beta = -15.94$ kg, $P = 1.4 \times 10^{-8}$). Follow-up analysis demonstrated that the observed signal is exclusively due to deletions (Supplementary Table 2). This CNV encompasses the BMI-associated lead SNP (rs6567160)[8, 28] (Fig. 3, Supplementary Table 8), but we observed virtually identical BMI-association *P* values for the SNP and the CNV in univariate and multivariate analysis. Hence, the two associations are most probably independent, further evidenced by the low LD between them ($r^2 = 0.0014$, $D' = 0.31$ in the UK Biobank). While rare height-increasing *MC4R* variants[27] have been previously reported, we found no height-effect of any CNV probes in this regions. Previous evidence for CNVs affecting BMI in the *MC4R* gene is scarce—there is only one case report of a 9-year-old obese boy carrying a larger (2.6 Mb) deletion encompassing the *MC4R* gene[21]. The top associations between this CNV and 27 tested traits in the UK Biobank are listed in Supplementary Tables 15 and 16.

**7q11.23 rearrangement.** The only WHR-specific genome-wide significant association implicated a CNV in the 7q11.23 region (72.6–73.58 Mb, Supplementary Figs. 11 and 12). Duplication carriers tend to have a higher waist-hip ratio ($\beta = 0.11$, $P = 1.5 \times 10^{-6}$). Separate analysis of deletions and duplications and absolute effect association did not show any stronger association, nevertheless, the effect of the duplication is slightly larger than that of the deletion (Supplementary Table 2). This CNV was recently found to be associated with schizophrenia in a large

case–control study[15]. It also overlaps with a 1.55–1.84 Mb long region known as the Williams–Beuren (WB) syndrome critical region (WBSCR)[29, 30]. WB syndrome[31, 32] is responsible for several complications: cardiovascular disease, neurologic abnormalities, attention deficit hyperactivity disorder, cognitive impairment, distinctive behavioral, and social traits. Due to selection bias, our prevalence estimation for the duplication (0.005%) and the deletion (0.005%) is somewhat lower than what is estimated for the WBSCR in the literature (0.005–0.013% for the duplication[33, 34] and 0.008–0.013% for the deletion[35]). The top associations between this CNV and 27 tested traits in the UK Biobank are listed in Supplementary Tables 17 and 18.

**3q29 rearrangement.** We discovered two CNVs in the 3q29 region, one 256 kb long (197.6–197.9 Mb), affecting weight ($\beta = 22.55$ kg, $P = 1.6 \times 10^{-6}$, deletion frequency = 0.004%, duplications frequency = 0.005%) and one 212 kb long (198.2–198.4 Mb) affecting height ($\beta = 13.3$ cm, $P = 2.3 \times 10^{-8}$, deletion frequency = 0.007%, duplications frequency = 0.007%) (Supplementary Figs. 13 and 14). Running the meta-analysis separately on the UKBB and the other cohorts, it appears that the signal comes mainly from the UK Biobank, however, without evidence for strong heterogeneity (Cochran $P > 0.05$). The proportional effects on height and weight are concordant with the fact that no association has been found with BMI or WHR. Children with this 3q29 deletion suffer from feeding problems, which may result in reduced adult weight[36]. A recurrent syndromic CNVs encompassing the two segments has recently been reported to be associated with schizophrenia[15]. On the anthropometric aspect, case reports from the literature are in agreement with our findings regarding the deletion impact on both weight and height[24, 37, 38]. Concerning the duplication, the phenotype spectrum is wider and the literature mainly reports obese/overweight cases, which is in agreement with our weight estimates. The reported effect on height is less pronounced. Our (median) deletion frequency

(0.004 and 0.007%, for the two segments respectively) is slightly over the reported value in a control population (0.003%)[36]. Finally, upon closer inspection of the region (Supplementary Fig. 14) we observed that the centromeric part of the CNV is implicated in weight regulation, while the telomeric end impacts height (implicated genes are listed in the legend of Table 1). Deletion and duplication frequencies were too low to be able to reliably establish the effects of deletions and duplications separately. The top associations between this CNV and 27 tested traits in the UK Biobank are listed in Supplementary Tables 19–22.

**CNVs with variable frequency across geographic locations**. Our meta-analysis revealed two population-specific CNVs. The first one is the CNV overlapping *FIBIN* and *BBOX1*, for which Finnish population cohorts have much higher deletion frequency. The second one, near *MC4R*, is specific for UK population cohorts. In both cases we compared potential confounding factors, such as probe densities, call rates, and CNV quality, but none of these could explain the frequency differences (Supplementary Figs. 8 and 10). Note that the frequency of the *MC4R* CNV both in the UK Biobank (0.028% (del), 0.005% (dup)) and in other UK cohorts genotyped on Illumina arrays (0.018% (del), 0.009% (dup)) is consistently higher than the frequency in non-UK samples (0.006% (del), 0.005% (dup)). Thus, the observed frequency difference is, at least in part, not due to array effect.

## Discussion

Our genome-wide CNV association meta-analysis on four anthropometric traits in ~190,000 unrelated adults showed a non-negligible CNV burden effect on BMI, height, and WHR. Furthermore, we identified seven CNVs significantly associated with at least one trait and three additional CNV regions have a close to genome-wide significant effect on one of the four traits (Supplementary Table 23). The analysis also gave new insights into the two 16p11.2 rearrangements[13, 14, 22].

As a proof of concept, we looked at CNVs known to be associated with BMI or obesity[39, 40]. Only one CNV (22q11.21) outside the 16p11.2 region was confirmed (Supplementary Fig. 15, Supplementary Tables 24 and 25). This difference might in part be explained by insufficient power or by the fact that, contrary to most previously published CNV studies[13, 15, 39, 41], our samples come from general populations. In addition, some of the previously reported CNVs might have been population-specific or simply spurious[42].

CNV burden analysis confirmed the already observed effect on BMI[19] and height[18], and showed an important effect on fat distribution (WHR). These observed signals are dominated by the deletions (up to five-fold larger effects), while the duplication effects are minor, except for WHR. The overall CNV burden has seemingly opposite effects on height and BMI, compatible with having no significant CNV burden effect on weight.

Overall, the genome-wide *P* values showed good adherence to the null distribution (Supplementary Fig. 16). For well-powered GWAS studies on heritable traits (e.g., height[9] and menarche[43]) high genomic control lambda value rather reflects true polygenic signals than uncorrected population stratification[44]. This was the case for our study too: while we observed inflated genomic lambda coefficients ($\lambda = 1.16$ (height), $\lambda = 1.12$ (weight), $\lambda = 1.08$ (BMI) and $\lambda = 1.05$ (WHR)), upon applying LD score regression[44] in the UK Biobank sample the intercept terms revealed no unaccounted population stratification ($\lambda_{LD} = 0.971$(height), $\lambda_{LD} = 1.005$(weight), $\lambda_{LD} = 0.993$ (BMI), $\lambda_{LD} = 0.942$ (WHR)).

Among the seven significant CNVs, two might be ancestry-specific, one Finnish and one British. It is not surprising, as these two populations have contributed the most samples to our meta-

analysis. These results show the need for collecting large population cohorts of the same origin since the frequency of many CNVs may vary across populations. Therefore, we believe that in the future, collecting larger, genotyped population-based cohorts from other countries and ethnicities could be an efficient way to discover novel trait-associated CNVs with larger effects.

Although we would have had higher power to detect associations with common CNVs, all of the anthropometric trait-associated CNVs identified in this study are rare (0.01–0.07%). This may be explained by the massive shift of CNV frequency spectrum compared to that of SNVs: based on CNV calls from >191,000 samples we observed that more than 92.4% of the CNVs are present in <1 in 1000 samples and 99.4% of them are rare (<1%). We are unsure whether the reason for the very low number of common CNVs is due to the detection technology or whether it reflects the nature of the underlying genomic events. Given the low frequency of most of the discovered CNVs and the neighboring genome structure, the majority of them may result from de novo and recent rearrangements. The total explained variance for all these rearrangements is ~0.09% for BMI, 0.10% for weight, 0.14% for height, and 0.04% for waist–hip ratio.

Our conditional analysis showed that CNV probes in the 16p11.2 region explain a substantial fraction of the association between all previously published SNPs near *SH2B1* and BMI and, similarly, the association between the SNP near *FLJ25404* and height. None of the remaining associated CNVs showed evidence for tagging common SNPs, nor do they explain known height/BMI-SNP associations. Note, however, that our CNV data are much noisier than SNP calls and thus the measured CNVs are poorer proxies for the true CNV status, which biases the conditional analysis towards the null (no tagging). Still, most of the obtained results are in line with the proposed theory that the majority of the discovered disease-associated common SNPs are not synthetic associations due to rare variant tagging[45].

Our study, besides reporting the association with anthropometric traits, can serve as an atlas of CNV maps based on a large general population of European ancestry[46, 47] (https://cnvcatalogue.bbmri.nl/ and underlying data in Supplementary Data 1). Similarly to large compendia of sequenced population individuals (e.g., EXaC[48]) for whole exome-/genome-sequence analysis for rare diseases, our inventory of CNV frequencies could help estimating their pathogenicity in the rare disease setting.

So far, many anthropometric GWASs have focused on BMI or height, but less on weight. In our analysis we found that studying the effect of CNVs on height and weight separately can carry important additional information beyond what we can learn from looking only at BMI. All the CNVs found to be associated with BMI were also associated either with height or weight, but the opposite does not hold. CNVs affecting height and weight in the same direction (e.g., 1q21.1) have less impact on BMI.

Our study has several weaknesses, which we tried to mitigate. Despite the fact that a plethora of software has been developed to detect CNVs from SNP array platforms, these genotyping chips were not initially designed for this purpose. This drawback reduces statistical power in our analysis by introducing false positive and false negative CNV calls. Importantly, there is no particular reason to believe that CNV calling artefacts appear specifically for samples enriched for low/high trait values. Thus, we believe that false CNV calls do not translate to false positive findings, but of course can substantially reduce statistical power. We did not perform independent (e.g., qPCR) experiments to confirm these CNV findings, but provided several lines of evidences to support our claims: (i) most of our reported CNVs have been reported before with similar frequency and breakpoints; (ii) many of our CNVs fall into regions already associated with obesity; (iii) QQ-plots for all traits show excellent adherence to

the null for the bulk of the CNV probes (Supplementary Fig. 16); (iv) our top CNVs show little or no heterogeneity across studies; (v) cohorts used 15 different genotyping arrays eliminating array-specific artefacts. Crucially, these genotyping platforms are more reliable than low-coverage sequencing to infer CNVs and thus remain the most cost-efficient to perform such large CNV-association studies. Another limitation is selection bias: As shown in the results, many of the anthropometric trait-associated CNVs we discovered are syndromic and were already observed in patients with specific genomic disorders (including traits like e.g., developmental delay[41], schizophrenia[15], etc.). In such situations we cannot distinguish whether the effect of those are mere consequence of the primary syndromes or trigger molecular mechanisms that act independently on anthropometric traits. Note, however, that this criticism is valid for any GWAS. The anthropometric effects of most of the syndromic CNVs are often poorly reported due to the small number of cases. We found evidence that 1q21.1 duplication carriers fall to the ~80th population height percentile[49], equivalent to ~4 cm of height increase, close to our observed effect of 3.46 cm. Carriers of the 22q11.2 deletion have on average 3 kg/m$^2$ higher BMI by the age of 20[50], which is comparable to our estimated effect of 4 kg/m$^2$. Moreover, our pheWAS analysis on 27 traits in the UK Biobank did not identify any non-anthropometric trait to be stronger associated with the discovered CNVs. Furthermore, we could not identify any trend between effects on anthropometric traits and schizophrenia (Supplementary Table 26), indicating that the anthropometric associations we observed cannot be secondary to schizophrenia. These lines of evidence indicate that most of the discovered CNVs affect anthropometric traits either primarily or in a disease-independent fashion. Importantly, the samples we analyzed come from population-based cohorts, healthier than the general population[51]. Selection bias, thus, removes many carriers of CNVs with larger effects[52], implying that the effects seen in our study are potentially smaller than the real ones. A further limitation is the variability in the frequency of such rare CNVs across populations. This phenomenon may render some of these discoveries difficult to replicate across populations, as not only similarly large replication studies would be necessary, but also populations in which the CNV frequency is high enough to yield sufficient statistical power. The final weakness to mention is that —to reduce cohort analyst burden—although we adjusted the analyzed traits for gender, but did not perform sex-specific analysis, which will be the central focus of a future study.

CNV studies in general population cohorts have been neglected in the past due to data availability issues. We have shown that such studies are feasible through a careful re-analysis of existing genotyping data. Our study has identified several height- and obesity-associated rare CNVs with substantial effect. We hope that our study will open new avenues for research to understand the impact of CNVs on human health on an unprecedented scale. The pipeline used for this meta-analysis could be applied with any other type of quantitative trait, and, with some modification, to any binary trait. Given the considerable overlap between CNVs associated with anthropometric traits, developmental delay and schizophrenia, in the future it would be insightful to switch point of view and apply a pheWAS approach in large, phenotype-rich cohorts such as the UK BioBank, allowing deeper interpretation of candidate CNVs.

## Methods

**Cohorts**. We conducted the meta-analysis for BMI (N UKBB = 119,873, N GIANT = 71,288), weight (N UKBB = 119,767, N GIANT = 55,416), height (N UKBB = 116,259, N GIANT = 65,706) and waist-hip ratio (N UKBB = 119,867, N GIANT = 41,377) (Supplementary Table 27). All GIANT samples were genotyped on Illumina platforms and the UK BioBank was genotyped on Affymetrix Axiom.

Participants of each cohort have signed the informed consent form of the respective study. In addition to the ethical committee approval of each individual study, this meta-analysis effort was also approved by the steering committee of the GIANT Consortium and the ethical review board of the UK Biobank (applications #17085, #16389, #9072). Only unrelated adults (genetic relatedness < 0.1) of European ancestry were included in the study.

**CNV calling**. For the Affymetrix Axiom chip, additional preprocessing was necessary: Raw probeset intensity values were quantile-normalized. Briefly, intensities were sorted numerically across each chromosome with missing values being allocated an overall median value to facilitate normalization. The mean intensity across each genotyping batch was then calculated for each sorted position. Mean intensities were then substituted in place of the equivalently ranked raw intensities whilst ignoring missing values. Each transformed intensity value was then log$_2$ transformed for processing in PennCNV-Affy. Mean values of all intensities across each chromosome were checked to ensure that they were the same within each genotyping batch of the UK Biobank. PennCNV-Affy was used to infer genotype clusters, generate Log R Ratio (LRR) and B-Allele Frequencies (BAF). All other cohorts used Illumina arrays, so LRR and BAF values were readily available.

We devised a pipeline that takes as input normalized BAF and LRR for each probe and sample (using the PennCNV software[53]), assigns probabilistic CNV calls[16] and runs association with each trait. The pipeline created a "population B allele frequency" (PFB) file for each cohort based on 200 randomly selected final reports. Adjacent CNVs with small gaps (gap shorter than 20% of the total length) were merged using the default PennCNV parameters. Finally, samples with more than 200 CNVs were excluded from further analysis.

For each CNV probe the pipeline computed a quality score (QS)[16]. The QS, based on the PennCNV quality metrics, estimates (Supplementary Table 28) the probability for a pennCNV call to be a true positive CNV call. It is a continuous value between −1 and 1, representing the product of the relative copy number (+1 for duplication, −1 for deletion) and the probability of the call being true, i.e., it is the expected copy number dosage relative to the copy neutral (2 copy) state. The QS is computed for each probe $j$ and sample $i$ (QS$_{i,j}$) and used as genotype value for CNV-trait association assuming a mirror effect of deletions and duplications. Since the QS accounts for various CNV characteristics (length, number of probes, etc.) we did not apply any filtering on these scores, which was shown to be the most powerful strategy for association[16]. However, probes with low imputation quality (see below) are filtered out in each cohort.

**CNV associations with anthropometric traits**. We focussed on the following anthropometric traits: BMI, weight, height and waist-hip ratio. BMI (kg/m$^2$), weight (kg), and waist-hip ratio were adjusted for sex, age, age$^2$ and the first five principal components of the genotype data when available. Height (in meters) was adjusted for sex, age and the first five principal components of the genotype data when available. The resulting trait residuals were then inverse normal quantile transformed.

As CNV boundaries vary across individuals, all associations were performed at the probe level. For this, CNV calls and quality measures were translated to probe level. For each probe in each cohort, association summary statistics are computed and collected for meta-analysis. The summary statistics for each probe are the mean QS, the sum of squared QS s, the sum of the phenotype–genotype products, the phenotype means and the sum of the squared phenotype values. These quantities are sufficient to compute regression coefficients as if we had access to each individual cohort data, which is, for rare variant associations, advantageous compared to standard inverse-variance meta-analysis.

**Summary statistics imputation**. As we collected different SNP arrays with variable probe content we imputed summary statistics to the Illumina 1M probes as reference probe set. For each Illumina 1M probe not present in the summary statistic probe list for a study we imputed its summary statistics based on the closest neighboring probes on each side within a 5 kb window. The imputation weights are set to be inversely proportional to the distance between the target probe and the neighboring probes.

**CNV imputation quality**. Analogously to genotype imputation, we used the MACH $\hat{r}^2$ measure[54] to estimate the quality of the CNVs estimation using the QS. This measure is the ratio of the variance of the Bernoulli distributed probabilistic CNV (taking value 1 with probability |QS$_{i,j}$|, 0 otherwise) averaged over the samples to the empirical variance of an expected dosage across all samples

$$\hat{r}_j^2 = \frac{\sum_i |QS_{i,j}| - QS_{i,j}^2}{\sum_i \left(|QS_{i,j}| - \overline{QS._j}\right)^2}$$

where QS$_{i,j}$ represents the QS for individual $i$ and probe $j$, and $\overline{QS._j}$ is the average QS$_{i,j}$ for probe $j$ across the samples. For each cohort, only probes with $\hat{r}_j^2 \geq 0.1$ were kept for the meta-analysis. Imputation qualities were also meta-analyzed using sample-size weighting.

**Pipeline**. Our published pipeline[16] based on bash, perl and R has been implemented to run all pre-meta-analysis steps. In brief, this pipeline formats the genotype files to run CNV calls using PennCNV, it cleans and merges the raw CNV calls, it computes a QS for each CNV and it finally calculates the summary statistics at the probe level. All participating cohorts ran the exact same pipeline and shared summary statistics with us. An example configuration file can be found in the Supplementary Note 1.

**Meta-analysis**. We ran fixed effects meta-analysis as described by RAREMETAL[55]. We directly computed the meta $\beta_{Meta}$ and $se_{Meta}$ for a given CNV probe from the summary statistics from the multiple cohorts:

$$\beta_{Meta} = \frac{\sum_c pg_c}{\sum_c g_c^2 - N_c * \overline{g_c}^2}$$

$$se_{Meta} = \sqrt{\frac{1}{\sum_c g_c^2 - N_c * \overline{g_c}^2}}$$

where $g_c^2$ is the sum of the squared CNV dosage for all individuals in cohort $c$, $pg_c$ is the sum of the products of phenotype × CNV dosage values for all individuals in cohort $c$, $\overline{g_c}$ is the average CNV dosage in cohort $c$, and $N_c$ is the sample size of cohort $c$. An overall Z score can be estimated as: $Z_{Meta} = \beta_{Meta}/se_{Meta}$. Eventually $P$ values are computed for each probe as: $P_{value} = 2 * \phi(-|Z_{Metal}|)$. All reported results in the paper are based on the full study population, unless stated otherwise (e.g., conditional analysis).

In order to decrease the number of tests and to avoid spurious associations we kept only probes that were CNV in at least four cohorts and had a frequency of at least 0.01%. Beside, using a 1 Mb sliding window over the entire genome, we merged probes with exactly the same summary statistics (frequency, effect size, SE), i.e., highly likely having the same profile across all individuals, within that window.

**Number of effective tests**. Based on an in-house cohort (HYPERGENES, $N = 2930$) we estimated the number of effective tests for the probes that were not discarded at the previous step ($N_{total} = 399,665$). We computed the QS for this cohort and kept only the probes for which the QS was not zero ($N_{non-zero}$). For each chromosome we used a 1 Mb window to calculate the number of effective tests locally. The number of effective tests corresponds to the number of probes explaining 99.5% of the variance in the window[17]. The results for each window and chromosome were then summed to obtain an overall $N_{eff}$ number of effective tests for the non-zero probes. This indicated the strength of dependence between CNV probes, $f = N_{eff}/N_{non-zero}$. We obtained a ratio of $f = 8.23$ and applied this scaling constant to the 242,022 probes tested in our meta-analysis study, yielding 29,407 independent tests and subsequently a $1.7 \times 10^{-6}$ genome-wide significant threshold. To ensure robustness, we repeated the same analysis for each of the 33 batches of the UK Biobank samples and obtained a slightly less stringent ratio (median $f = 13.83$, $CI_{95\%} = [8.20, 20.17]$), but we preferred to use the more conservative threshold of $1.7 \times 10^{-6}$.

**CNV burden analysis**. For each sample we calculated the total number of (imputed) CNV probes showing deviation from the copy neutral state. To account for uncertainty in the calls and to avoid arbitrary thresholding, we used the absolute QS of each probe (for the U-shaped model) and summed them up for the whole genome. For the other models (deletion-, duplication-only) we used the respective modifications (minus deletion QS, duplication QS). We then ran a linear regression between the total burden score and the various traits and meta-analyzed the results from the 26 cohorts.

**Candidate CNV regions**. In order to validate our methodology, we decided to first look at regions already associated with BMI, weight or height. Regions have been defined based on proximity of GWAS SNP[8, 9], CNVs report[39], genes from OMIM repository[15] and from a very recent systematic review of known genes implicated in genetic syndromes with obesity (Table 1 of Kaur et al.[40]). Regarding the candidate CNVs/genes or the OMIM regions, all high quality ($r^2 > 0.5$) probes falling into the regions were selected. For each GWAS SNP we selected all the probes with association results in a $\pm500$ kb region around the SNP position. The CNV report cataloged 84 BMI and obesity-associated CNVs from research published since 2008 via PubMed search (see Supplementary Table 2 of the publication by Petersen et al.[39]). Out of the 84 CNVs, we had good quality probes for 48 of them that we subsequently tested. Out of the 79 OMIM regions for weight and BMI, 37 had good quality probes ($r^2 > 0.5$). The 96 Kaur et al. genes[40] represent 65 regions, out of which 57 are on autosomes and 32 of those contained probes within 10 kb with good imputation quality ($r^2 > 0.5$). In every candidate region we computed the minimal $P$ value and multiplied it with the number of effective tests for that region to obtain one (corrected) $P$ value per region. We then computed quantile–quantile plot to visually inspect potential inflation and computed the fold-enrichment of regions with low $P$ values ($P < 0.05$).

**GWAS and eQTL lookup**. In order to further interpret our findings, we checked whether height-associated coding variants[10] were located within our height-associated CNVs. Genes whose expression is modulated by both trait-associated SNPs and CNVs are good gene candidates and can help narrowing down the critical region. To identify such genes, we asked whether known height/BMI-associated SNPs[8, 9] act also as *cis* eQTLs in blood[56] for the genes located within height/BMI-associated CNVs.

**Data availability**. All association results are available in Supplementary Data 1 and can also be browsed at https://cnvcatalogue.bbmri.nl

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

## Acknowledgements

This research has been conducted using the UK Biobank Resource. This research has been conducted using the Danish National Biobank resource. The authors are grateful to the Raine Study participants and their families, and to the Raine Study research staff for cohort co-ordination and data collection. QIMR is grateful to the twins and their families for their generous participation in these studies. We would like to thank staff at the Queensland Institute of Medical Research: Anjali Henders, Dixie Statham, Lisa Bowdler, Ann Eldridge, and Marlene Grace for sample collection, processing and genotyping, Scott Gordon, Brian McEvoy, Belinda Cornes and Beben Benyamin for data QC and pre-paration, and David Smyth and Harry Beeby for IT support. HBCS Acknowledgements: We thank all study participants as well as everybody involved in the Helsinki Birth Cohort Study. Helsinki Birth Cohort Study has been supported by grants from the Academy of Finland, the Finnish Diabetes Research Society, Folkhälsan Research Foundation, Novo Nordisk Foundation, Finska Läkaresällskapet, Juho Vainio Founda-tion, Signe and Ane Gyllenberg Foundation, University of Helsinki, Ministry of Edu-cation, Ahokas Foundation, Emil Aaltonen Foundation. Finrisk study is grateful for the THL DNA laboratory for its skillful work to produce the DNA samples used in this study and thanks the Sanger Institute and FIMM genotyping facilities for genotyping the samples. We thank the MOLGENIS team and Genomics Coordination Center of the University Medical Center Groningen for software development and data management, in particular Marieke Bijlsma and Edith Adriaanse. This work was supported by the Leenards Foundation (to Z.K.), the Swiss National Science Foundation (31003A_169929 to Z.K., Sinergia grant CRSII33-133044 to AR), Simons Foundation (SFARI274424 to AR) and SystemsX.ch (51RTP0_151019 to Z.K.). A.R.W., H.Y. and T.M.F. are supported by the European Research Council grant: 323195:SZ-245. M.A.T., M.N.W. and An.M. are supported by the Wellcome Trust Institutional Strategic Support Award (WT097835MF). For full funding information of all participating cohorts see Supple-mentary Note 2.

## Author contributions

Z.K. and T.M.F. designed the study, Au.M. and Z.K. designed the analysis pipeline, Au. M. analyzed, QC'ed the data and performed the meta-analysis, Au.M. and Z.K. con-ducted all follow-up analyses. Au.M., T.M.F., and Z.K. wrote the paper. A.B.N., A.L., An . M., A.R.W., A.S.H., C.H., E.P., E.P.B., E.V., F.G., F.R., G.W.M., H.Y., I.M.H., I.M.N., J.C. C., J.K., K.S.R., M.A.T., M.F.F., M.K.W., M.M., M.N., M.P., N.S., Pan.D., Pat .D., P.L., R. M.F., S.A., S.E.J., T.M., T.T.P., T.W.W., U.S., V.S., and W.Z. contributed to individual study design and management; A.B.N., A.L., An.M., Au.M., A.R.W., A.S.H., C.P., E.P.B., E.S., E.V., G.W.M., H.B., H.Y., I.M.H., J.C.C., J.K., J.S., J.S.K., J.T., M.A.P., M.F.F., M.L., M.N., M.P., N.G.M., N.S., Pan.D., Pat.D., P.L., R.M., R.M.F., S.A., S.E.J., S.M., T.M., T.M. F., T.T.P., T.W.W., U.S., W.Z. played a role in data collection; A.B.N., Au.M., A.P., C.H., C.P., F.G., G.W.M., I.M.H., J.C.C., M.A.P., M.M., M.N., M.N.W., M.P., N.S., Pan.D., P.L., R.M., R.M.F., S.E.J., W.Z. contributed to the genotyping; A.B.N., Au.M., A.R.W., A.S.H., E.S., F.G., G.W.M., I.M.H., J.K., J.T., M.F.F., M.L., M.N., M.N.W., M.P., N.G.M., Pat.D., P.K., R.N.B., S.E.J., V.S., W.Z. participated in the phenotype preparation; A.B.N., Au.M., B.F., E.S., L.F., M.L., M.N., M.N.W., N.R.W., Pat.D., P.J.V., R.M.F., R.N.B., Y.S. performed study specific analysis; A.J.O., Au.M., C.V., D.C., D.P., D.P.S., D.R.N., G.L., H.S., K.B., K. C., K.C., K.K., K.M., M.A.S., M.A.T., M.F.F., M.K.W., M.M., M.N., M.N.W., N.S., R.J.F.L., S.E.M., T.D.S., T.T.P., W.A., X.L., Z.K. contributed to/supervised study specific analysis.

## Additional information

**Competing interests:** The authors declare no competing financial interests.

Aurélien Macé[1,2,3], Marcus A. Tuke[4], Patrick Deelen[5,6], Kati Kristiansson[7,8], Hannele Mattsson[7,8], Margit Nõukas[9,10], Yadav Sapkota[11,12], Ursula Schick[13], Eleonora Porcu[2,14], Sina Rüeger[1,2], Aaron F. McDaid[1,2], David Porteous[15], Thomas W. Winkler[16], Erika Salvi[17], Nick Shrine[18], Xueping Liu[19], Wei Q. Ang[20], Weihua Zhang[21,22], Mary F. Feitosa[23], Cristina Venturini[24], Peter J. van der Most[25], Anders Rosengren[26,27], Andrew R. Wood[4], Robin N. Beaumont[4], Samuel E. Jones[4], Katherine S. Ruth[4], Hanieh Yaghootkar[4], Jessica Tyrrell[4], Aki S. Havulinna[7], Harmen Boers[5,6], Reedik Mägi[9], Jennifer Kriebel[28,29,30], Martina Müller-Nurasyid[31,32,33], Markus Perola[7,34], Markku Nieminen[35], Marja-Liisa Lokki[36], Mika Kähönen[37,38], Jorma S. Viikari[39,40], Frank Geller[19], Jari Lahti[41,42], Aarno Palotie[8,43,44], Päivikki Koponen[7], Annamari Lundqvist[7], Harri Rissanen[7], Erwin P. Bottinger[13], Saima Afaq[21], Mary K. Wojczynski[23], Petra Lenzini[23], Ilja M. Nolte[25], Thomas Sparsø[26,27], Nicole Schupf[45], Kaare Christensen[46], Thomas T. Perls[47], Anne B. Newman[48], Thomas Werge[26,27,49], Harold Snieder[25], Timothy D. Spector[24], John C. Chambers[21,22,50], Seppo Koskinen[7], Mads Melbye[19,51,52], Olli T. Raitakari[53,54], Terho Lehtimäki[55,56], Martin D. Tobin[18,57], Louise V. Wain[18,57], Juha Sinisalo[35], Annette Peters[29,30,33], Thomas Meitinger[58,59], Nicholas G. Martin[60], Naomi R. Wray[61], Grant W. Montgomery[11,62], Sarah E. Medland[11], Morris A. Swertz[5,6], Erkki Vartiainen[7], Katja Borodulin[7], Satu Männistö[7], Anna Murray[4], Murielle Bochud[1], Sébastien Jacquemont[63,64], Fernando Rivadeneira[65,66], Thomas F. Hansen[26,27], Albertine J. Oldehinkel[67], Massimo Mangino[24,68], Michael A. Province[23], Panos Deloukas[69,70], Jaspal S. Kooner[22,50,71], Rachel M. Freathy[4], Craig Pennell[20], Bjarke Feenstra[19], David P. Strachan[72], Guillaume Lettre[73,74], Joel Hirschhorn[75,76,77], Daniele Cusi[17,78], Iris M. Heid[16], Caroline Hayward[79], Katrin Männik[9,14], Jacques S. Beckmann[2], Ruth J.F. Loos[13,80], Dale R. Nyholt[11,81], Andres Metspalu[9], Johan G. Eriksson[82,83], Michael N. Weedon[4], Veikko Salomaa[7], Lude Franke[5], Alexandre Reymond[14], Timothy M. Frayling[4] & Zoltán Kutalik[1,2]

[1]Institute of Social and Preventive Medicine, Lausanne University Hospital, Lausanne 1010, Switzerland. [2]Swiss Institute of Bioinformatics, Lausanne 1015, Switzerland. [3]Department of Computational Biology, University of Lausanne, Lausanne 1011, Switzerland. [4]Genetics of Complex Traits, University of Exeter Medical School, University of Exeter, Exeter EX2 5DW, UK. [5]Department of Genetics, University of Groningen, University Medical Center Groningen, Groningen 9713 GZ, The Netherlands. [6]University of Groningen, University Medical Center Groningen, Genomics Coordination Center, Groningen 9713 GZ, The Netherlands. [7]National Institute for Health and Welfare, Helsinki 00271, Finland. [8]Institute for Molecular Medicine Finland, University of Helsinki, Helsinki FI-00014, Finland. [9]Estonian Genome Center, University of Tartu, Tartu 51010, Estonia. [10]Institute of Molecular and Cell Biology, University of Tartu, Tartu 51010, Estonia. [11]QIMR Berghofer Medical Research Institute, Brisbane 4006, Australia. [12]St. Jude Children's Research hospital, Memphis, TN 38105, USA. [13]The Charles Bronfman Institute for Personalized Medicine, Icahn School of Medicine at Mount Sinai, New York, NY 10029, USA. [14]Center for Integrative Genomics, University of Lausanne, Lausanne 1015, Switzerland. [15]Generation Scotland, Centre for Genomic and Experimental Medicine, Institute of Genetics and Molecular Medicine, University of Edinburgh, Edinburgh EH4 2XU, UK. [16]Department of Genetic Epidemiology, University of Regensburg, Regensburg 93053, Germany. [17]Department of Health Sciences, University of Milano, Milano 20090, Italy. [18]Department of Health Sciences, University of Leicester, Leicester LE1 7RH, UK. [19]Department of Epidemiology Research, Statens Serum Institut, Copenhagen 2300, Denmark. [20]School of Women's and Infants' Health, The University of Western Australia, Perth 6009, Australia. [21]Department of Epidemiology and Biostatistics, School of Public Health, Imperial College London, Norfolk Place, London W2 1PG, UK. [22]Department of Cardiology, Ealing Hospital NHS Trust, Uxbridge Road, Southall, Middlesex UB1 3EU, UK. [23]Department of Genetics, Washington University School of Medicine, St. Louis 63108, USA. [24]Department of Twin Research and Genetic Epidemiology, King's College London, London SE1 7EH, UK. [25]Department of Epidemiology, University of Groningen, University Medical Center Groningen, Groningen 9713 GZ, The Netherlands. [26]Research Institute of Biological Psychiatry, Mental Health Center Sct. Hans, Roskilde 4000, Denmark. [27]iPSYCH, The Lundbeck Foundation Initiative for Integrative Psychiatric Research, Aarhus 8210, Denmark. [28]Research Unit of Molecular Epidemiology, Helmholtz Zentrum Muenchen, German Research Center for Environmental Health, Neuherberg 85764, Germany. [29]Institute of Epidemiology II, Helmholtz Zentrum Muenchen, German Research Center for Environmental Health, Neuherberg 85764, Germany. [30]German Center for Diabetes Research (DZD), Muenchen-Neuherberg 85764, Germany. [31]Institute of Genetic Epidemiology, Helmholtz Zentrum München —German Research Center for Environmental Health, Neuherberg 85764, Germany. [32]Department of Medicine I, Ludwig-Maximilians-Universität, Munich 81337, Germany. [33]DZHK (German Centre for Cardiovascular Research), Partner Site Munich Heart Alliance, Munich 80336, Germany. [34]University of Tartu, Estonian Genome Center, Tartu 51010, Estonia. [35]Heart and Lung Centre HUCH and Helsinki University, Helsinki 00029, Finland. [36]Medicum, Helsinki University, Helsinki 00014, Finland. [37]Department of Clinical Physiology, Tampere University Hospital, Tampere 33521, Finland. [38]Department of Clinical Physiology, University of Tampere Faculty of Medicine and Life Sciences, Tampere 33521, Finland. [39]Division of Medicine, Turku University Hospital, Turku 20521, Finland. [40]Department of Medicine, University of Turku, Turku 20521, Finland. [41]Department of Psychology and Logopedics, Faculty of Medicine, University of Helsinki, Helsinki 00014, Finland. [42]Helsinki Collegium for Advanced Studies, University of Helsinki, Helsinki 00014, Finland. [43]The Stanley Center for Psychiatric Research, The Broad Institute of MIT and Harvard, Cambridge, MA 02142, USA. [44]Analytic and Translational Genetics Unit, Department of Medicine and Psychiatric & Neurodevelopmental Genetics Unit, Massachusetts General Hospital, Boston, MA 02114, USA. [45]Taub Institute, College of Physicians and Surgeons, Columbia University, New York, NY 10032, USA. [46]The Danish Aging Research Center, Epidemiology, University of Southern Denmark, Odense 5000,

Denmark. [47]Section of Geriatrics, Department of Medicine, Boston University, Boston School of Medicine and Boston Medical Center, Boston, MA 02118, USA. [48]Department of Epidemiology, University of Pittsburgh Graduate School of Public Health, Pittsburgh, PA 15261, USA. [49]Institute of Clinical Sciences, Faculty of Medicine and Health Sciences, University of Copenhagen, Copenhagen 2200, Denmark. [50]Imperial College Healthcare NHS Trust, London W12 0HS, UK. [51]Department of Clinical Medicine, University of Copenhagen, Copenhagen 2200, Denmark. [52]Department of Medicine, Stanford University School of Medicine, Stanford, CA 94305, USA. [53]Department of Clinical Physiology and Nuclear Medicine, Turku University Hospital, Turku 20521, Finland. [54]Research Center of Applied and Preventive Cardiovascular Medicine, University of Turku, Turku 20521, Finland. [55]Department of Clinical Chemistry, Fimlab Laboratories, Tampere 33521, Finland. [56]Department of Clinical Chemistry, University of Tampere Faculty of Medicine and Life Sciences, Tampere 33521, Finland. [57]National Institute for Health Research, Leicester Respiratory Biomedical Research Unit, Glenfield Hospital, Leicester LE3 9QP, UK. [58]Institute of Human Genetics, Helmholtz Zentrum München, Neuherberg 85764, Germany. [59]Institute of Human Genetics, Technische Universität München, Munich 81675, Germany. [60]QIMR Berghofer Medical Research Institute, Brisbane, QLD 4006, Australia. [61]Queensland Brain Institute, University of Queensland, Brisbane, QLD 4072, Australia. [62]The Institute for Molecular Bioscience, University of Queensland, Brisbane, QLD 4072, Australia. [63]Centre de recherche, Centre Hospitalier Universitaire Sainte Justine, Montréal, Canada H3T 1C5. [64]Service of Medical Genetics, University Hospital of Lausanne, University of Lausanne, Lausanne 1011, Switzerland. [65]Department of Epidemiology, Erasmus Medical Center, Rotterdam 3015 GE, The Netherlands. [66]Department of Internal Medicine, Erasmus Medical Center, Rotterdam 3015 GE, The Netherlands. [67]Department of Psychiatry, University of Groningen, University Medical Center Groningen, Groningen 9713 GZ, The Netherlands. [68]NIHR Biomedical Research Centre at Guy's and St Thomas' Foundation Trust, London SE1 9RT, UK. [69]William Harvey Research Institute, Barts and The London School of Medicine and Dentistry, Queen Mary University of London, London EC1M 6BQ, UK. [70]Princess Al-Jawhara Al-Brahim Centre of Excellence in Research of Hereditary Disorders (PACER-HD), King Abdulaziz University, Jeddah 21589, Saudi Arabia. [71]National Heart and Lung Institute, Imperial College London, Hammersmith Hospital Campus, Du Cane Road, London W12 0NN, UK. [72]Population Health Research Institute, St. George's, University of London, London SW17 0RE, UK. [73]Montreal Heart Institute, Université de Montréal, Montreal, QC, Canada H1T 1C8. [74]Department of Medicine, Faculty of Medicine, Université de Montréal, Montreal, QC, Canada H3T 1J4. [75]Broad Institute of MIT and Harvard, Cambridge, MA 02142, USA. [76]Division of Endocrinology, Boston Children's Hospital, Harvard Medical School, Boston, MA 02115, USA. [77]Department of Genetics, Harvard Medical School, Boston, MA 02115, USA. [78]Institute of Biomedical Technologies, Italian National Research Council, Milano 20090, Italy. [79]Medical Research Council Human Genetics Unit, Institute of Genetics and Molecular Medicine, University of Edinburgh, Edinburgh EH4 2XU, UK. [80]The Genetics of Obesity and Related Metabolic Traits Program, Icahn School of Medicine at Mount Sinai, New York 10029, USA. [81]Institute of Health and Biomedical Innovation, Queensland University of Technology, Brisbane 4059, Australia. [82]Department of General Practice and Primary Health Care, University of Helsinki and Helsinki University Hospital, Helsinki 00014, Finland. [83]Folkhälsan Research Center, Helsinki 00250, Finland. Timothy M. Frayling and Zoltán Kutalik jointly supervised this work.

