## [Peer Review File · Nature Communications]

Reviewers' comments:

Reviewer #1 (Remarks to the Author):

In this study, A. Mace and colleagues analyzed the association of CNVs with four anthropometric traits in up to 191,161 adults of European descent. They identified five novel rare CNV-associations and confirmed two previous ones. This is an interesting study.

1-P1: please mention that the participants are of European descent in the title.

2-P3: the authors mention that the GWAS BMI and height SNPs explain only 2.7% and 20% of the variance of respective traits in order to justify the study of CNVs that may 'fill the gap of missing heritability of complex traits'. The authors focus on GWAS significant SNPs, but recent polygenic studies confirm that SNPs may account for 60-70% and 30-40% of heritability for height and BMI, respectively (Yang, Nat Genet 2015). This is not surprising as SNPs represent the majority of genetic variants. On the other side, CNVs are unlikely to explain the missing heritability for complex traits. The authors are encouraged to provide a more balanced and mature introduction, stating that the heritability of complex traits is heterogeneous (SNPs, rare mutations, common and rare CNVs, major chromosomal abnormalities, GxG and GxE interactions, epigenetics....) and more CNV studies are needed. The authors may also provide an estimation of the % of variance of the four traits explained by the CNVs identified or confirmed in this study.

3-P3: the authors mention that rare and low frequency single nucleotide variants do not seem to explain a large fraction of missing heritability for height and BMI. However, only the rare variant height GIANT paper has been published so far (Marouli et al., Nature 2017), and the BMI exome chip paper is still in preparation. Please update the sentence accordingly.

4-P3: when you refer to the 600kb 16p11.2 deletion please cite the two seminal papers (Walters Nature 2010, Bochukova Nature 2010). The annotations BP2-BP3 and BP4-BP5 do not make sense for most readers, please explain these abbreviations or remove them.

5-P4: the authors used a mirror effect model (opposite and equal sized effect of deletions and duplications at any given locus) in the genome-wide CNV association study. Focusing on this model only is very restrictive and this represents a major limitation of the study. For instance, the 16p11.2 600kb deletions and duplications show mirror effects for BMI (deletion carriers are obese and duplication carriers are underweight), but both deletion and duplication carriers show psychiatric features (U-shaped model). The authors may test different models (mirror model, U-shaped model, effect only for deletions, effect only for duplications).

6-P4: 'For burden analysis all four models were tested'. Please provide more explanations on the models.

7-P4: please provide more details about the ref 20 study. The readers may not understand why you test these 59 BMI-associated CNVs.

8-P5: please refer to the recent systematic review on syndromic obesity by Kaur et al. (Obesity Reviews 2017) to identify syndromic obesity genes. Screening the OMIM repository alone does not provide an exhaustive list of syndromic obesity candidate genes.

9-The synthetic association hypothesis states that partial linkage disequilibrium between a rare variant conferring a strong genetic effect on a complex trait and a common variant may translate into a modest genetic effect of the SNP for the same trait. Considering that several BMI-associated rare CNVs in this study are located in well-established BMI-associated SNP loci (SH2B1, MC4R, FMO5, FLJ25404), it appears essential to exclude the possibility of synthetic association. The authors sometimes provide linkage disequilibrium information between the lead SNPs and CNVs,

but it is not sufficient. Even a modest linkage disequilibrium may explain the SNP association. The authors may perform additional analyses (e.g. conditional analyses) to formally demonstrate the independence of rare CNVs and common SNPs associations at these loci.

10-P5: the authors suggest that the association between the 600kb 16p11.2 deletion and BMI was mainly driven by its association with decreased height. It is a bold statement if we consider that 16p11.2 deletion carriers develop a highly penetrant form of monogenic hyperphagic obesity (Walters Nature 2010, Bochukova Nature 2010). Mentioning that the 600kb 16p11.2 deletion was associated with increased BMI and weight and decreased height may be sufficient.

11-P6: it is not clear if the association results in the UK Biobank are reported separately or are pooled with other GIANT studies.

12-I think that the associations between CNVs and 23-27 phenotypic traits in UK Biobank do not add much to the study. It is unclear if the authors applied a Bonferroni corrections for all the statistical tests performed in UK Biobank, and they do not provide a detailed description of the results. The authors are encouraged to focus on the analysis of the four anthropometric traits in UK Biobank (height, weight, BMI, waist-to-hip ratio), as a majority of other traits (e.g. forced expiratory volume, hand grip strength) are out of context.

13-P6: the cis eQTL analysis is out of context, please remove this section.

14-P7: please double-check that all the references are in superscript, and that the font size is unique. When you refer to the SNP near MC4R associated with BMI, please cite Loos et al., Nat Genet 2012.

15-P7: it is surprising that the CNV near MC4R is not associated with height (rare mutations and SNPs in/near MC4R are, see Loos et al., Nat Genet 2012). This deserves more discussion.

16-The authors mention that a majority of SNPs associated with anthropometric traits are also associated with schizophrenia in literature. This deserves more discussion.

17-P8: please remove the section 'Associations with suggestive significance'. These associations do not pass the multiple testing adjusted association threshold and are likely to be false-positive.

18-P9: the lack of replication of previous CNV associations can also signify that the associations reported in literature are spurious (please cite Walters et al., PLOS One 2013). Please update the text accordingly.

19 -P9: 'We believe that instead of collecting more samples from the UK and Finland, collecting larger population-based cohorts from other countries could be a more efficient way to discover novel trait-associated CNVs with larger effects'. Please update this over simplistic statement. First, these actions are not mutually exclusive. Second, genotyping genome-wide DNA arrays is more challenging and expensive than collecting biological samples in large populations. Third, the authors may also mention to perform CNV association studies in other ethnic groups.

20-P9: 'None of the associated CNVs seems to tag common SNPs, nor explain known height/BMI associations'. This is an overstatement. The data presented in the Results section do not enable the authors to make any conclusion.

21-P9: please replace 'an atlas of CNV map in the general population' by 'an atlas of CNV map in the general European ancestry population'.

Reviewer #2 (Remarks to the Author):

Summary

CNVs on anthro traits in ~191k individuals from 26 cohorts identified 5 novel associations and 2 known ones. The discovered associated CNVs are rare, and have large effect sizes. They used PENNCNV on Illumina or Affy Axiom to generate CNV calls and defined the threshold of significance to be 1.7×10^{-6} .

Major concerns

I wonder about the utility of doing a genome-wide association study of CNVs but using the more liberal study-wide threshold for significance of 1.7×10^{-6} . If the CNV is tagging a single nucleotide variant, it would not be described in a GWAS but would be here.

Definitely delete the section on "Associations with suggestive significance"

How can you really tell between population-specific CNVs and differences in CNV calling due to array type? particularly for the UK biobank? This explanation at the end of the results section is not convincing.

There is no discussion of which region(s) in the deletions might be functional. Any genes? Any critical regulatory elements?

The deletion burden analysis needs more detail.

The QQ plots look inflated - what were the lambdas for the CNV association tests?

I wonder if so many non-significant association results in Table 1 is useful. Why not make this a supplemental table and just give the genome-wide (or study-wide) significant results only with trait listed?

I would delete the section on candidate CNVs - not significant.

Minor concerns

"which share genetic basis with developmental- and psychiatric disorders." not sure if this implies the genes are in common between anthro and psych disorders or the genetic architecture is similar.

"We did not observe any CNV burden effect on the weight" what weight? Weight the trait?

59 BMI-associated CNVs manuscript 20? Are these previously reported at genome-wide significance?

Grammatical errors "Further three CNVs" and "Burden of rare CNV" "The first, 220kb BP2-BP3 deletion" etc.

"And finally one deletion, encompassing 204 BBOX1 and FIBIN genes, seems to be particularly frequent in the Finnish population (0.89% vs 0.02% in 205 the non-Finnish cohorts)." Is it associated with anything?

We could not narrow down the BMI association signal to the previously proposed SH2B1 (lowest $P=7.7 \times 10^{-8}$) as it spreads out to other genes, including SPNS1 and LAT (lowest $P=5.3 \times 10^{-8}$ 230)

(Figure 3) - please revise "as it spreads out"??

Reviewer #3 (Remarks to the Author):

The manuscript by Mace, Kutalik and colleagues entitled "CNV-association meta-analysis in 191,161 adults reveals new loci associated with anthropometric traits" reports the largest ever study of copy number variants (CNVs) with a complex trait. Taking advantage of height, weight and waist-hip ratio measurements from large cohort studies the authors have carried out a systematic analysis of the effects of CNVs on these anthropometric traits. Given that most CNVs that are known to influence height and weight are quite rare in the general population, it would be an accomplishment simply to identify a few credible associations. The sample size of this study, however, provided sufficient power to demonstrate a significant influence of CNV burden on height and weight and to identify seven loci that survive genome-wide multiple testing correction. These rare variants have large effects on height and weight (and neurodevelopment), providing further evidence that physical and cognitive traits have a shared genetic basis. Rare variants were identified at 3 loci that have been identified in previous GWAS (MC4R, FIBIN, FMO5), suggesting that common and rare variants of these genes influence these traits.

The data processing and statistical methods are highly rigorous and incorporate a number of clever innovations. The evidence supporting the association of CNVs with anthropometric traits is strong. I have several minor concerns to be addressed.

1. Data availability. After logging in to the website (<https://molgenis59.gcc.rug.nl>), the browser displays only genes, repeats and conservation tracks. The track labeled "CNVs" is empty. This needs to be fixed. Will a bulk download of the CNV calls also be available?
2. Line 107: There are few examples of robust associations between rare copy number variants and complex traits. Do you mean "anthropometric traits". In addition to many chromosomal rearrangements that are associated with Mendelian disease, there are dozens that are associated neuropsychiatric disorders (which qualify as "complex").
3. CNV calling: methods do not state whether any filters were applied on quality score, CNV length or number of probes. Such post-processing steps are typically done. If no such steps were taken, this should be clarified
4. If quality control was handled exclusively within the statistical model, by using the quality score as a genotype value, this represents an innovation that departs somewhat from the norm. Some additional attention to this would be helpful. For instance, If there are many low-quality calls overlapping a probe, the MACH r-squared value will approach zero and that probe will be filtered out. Correct? Assuming, the probe passes this filter, then how is the dosage of individuals in the cohort (G_c) calculated? How is the quality score used to calculate a weighted genotype value?
5. The basic statistical meta-analysis approach makes sense, but I am curious how this approach to filtering and meta-analysis influence the association results. As the allele frequencies become very high ($MAF > 5\%$), are these probes more likely to be filtered out? Or are common CNVs well represented?
6. Related to the previous question: One of the more puzzling results of this study is the LACK of association of common CNVs. The logical conclusion of this is that the effects are extremely modest. This is surprising though considering that common SNP associations are readily detectable in the same sample. Are common CNVs well ascertained? Well represented (see previous comment)? Perhaps there are just so few common alleles detectable with genotyping arrays?
7. Line 393: We believe that False CNV calls do not translate into false positive findings. Probably not, that's the beauty of a "case-only" design. Issues of poor data quality may affect specific dataset, but it would be unlikely for false CNV calls in the sample to correlate with clinical measures. In this kind of cohort study, CNV calling error would not lead to false-positive associations BUT it could diminish the evidence for association. This is something that should be considered when interpreting some of the negative results (lack of association of common CNVs or lack of replication of previous CNV findings.)
8. Supplementary Figures: Frequency units are %?

Reviewers' comments:

Reviewer #1 (Remarks to the Author)

In this study, A. Mace and colleagues analyzed the association of CNVs with four anthropometric traits in up to 191,161 adults of European descent. They identified five novel rare CNV-associations and confirmed two previous ones. This is an interesting study.

Thank you for the comment.

1-P1: please mention that the participants are of European descent in the title.

Good point, we added it now.

2-P3: the authors mention that the GWAS BMI and height SNPs explain only 2.7% and 20% of the variance of respective traits in order to justify the study of CNVs that may 'fill the gap of missing heritability of complex traits'. The authors focus on GWAS significant SNPs, but recent polygenic studies confirm that SNPs may account for 60-70% and 30-40% of heritability for height and BMI, respectively (Yang, Nat Genet 2015). This is not surprising as SNPs represent the majority of genetic variants. On the other side, CNVs are unlikely to explain the missing heritability for complex traits. The authors are encouraged to provide a more balanced and mature introduction, stating that the heritability of complex traits is heterogeneous (SNPs, rare mutations, common and rare CNVs, major chromosomal abnormalities, GxG and GxE interactions, epigenetics....) and more CNV studies are needed. The authors may also provide an estimation of the % of variance of the four traits explained by the CNVs identified or confirmed in this study.

We provide now a more balanced introduction including the points of the reviewer:

"Findings from the largest genome-wide association studies (GWAS), including over 250,000 samples, on BMI¹ and height² revealed 97 and 697 Single Nucleotide Polymorphisms (SNPs) explaining cumulatively only 2.7% and 20% of the variance of the respective phenotypes. Using genotyping arrays enriched for coding regions (exome-chip) large meta-analysis GWAS for height and BMI discovered several rare frequency coding single nucleotide variants (SNVs) associated with these traits, still these SNVs have thus far explained only a very small variation in these traits (e.g. 0.51% explained height variance³, and even smaller fractions are likely for BMI and WHR – personal communication). Nevertheless, random effect models accounting for imperfect imputation estimate that the total additive effect of all SNVs explain 56% and 27% of height and BMI variability, respectively⁴. While there is a growing consensus that predominantly SNVs contribute to the heritability, the impact of the structural architecture of the genome (copy number variants, complex rearrangements, etc.) is understudied and not negligible⁵. It has been shown that rare and large Copy Number Variants (CNVs), such as the 600 kb BP4-BP5 16p11.2 rearrangement^{6,7}, can exert substantial impact on BMI, but little effort has been made towards assessing the genome-wide impact of CNVs on complex traits. To our knowledge, only one genome-wide CNV-association study (on schizophrenia) has been performed in large adult population samples⁸."

The variance explained by all identified/confirmed CNVs was mentioned in the Discussion and estimated to be ~0.1%.

3-P3: the authors mention that rare and low frequency single nucleotide variants do not seem to explain a large fraction of missing heritability for height and BMI. However, only the rare variant height GIANT paper has been published so far (Marouli et al., Nature 2017), and the BMI exome chip paper is still in preparation. Please update the sentence accordingly.

We did not get the permission from the BMI and waist exome-chip PIs to release the exact numbers before those papers are published, but are authorized to allude to the fact that those numbers are even smaller than those for height. We added this information to the Introduction.

4-P3: when you refer to the 600kb 16p11.2 deletion please cite the two seminal papers (Walters Nature 2010, Bochukova Nature 2010). The annotations BP2-BP3 and BP4-BP5 do not make sense for most readers, please explain these abbreviations or remove them.

We added now to the text that BP stands for breakpoint and the corresponding references (13&14) describe the definition in full details. We believe that specialists appreciate this extra precision and for less specialized readers hopefully it is not too confusing now with the extra explanation.

5-P4: the authors used a mirror effect model (opposite and equal sized effect of deletions and duplications at any given locus) in the genome-wide CNV association study. Focusing on this model only is very restrictive and this represents a major limitation of the study. For instance, the 16p11.2 600kb deletions and duplications show mirror effects for BMI (deletion carriers are obese and duplication carriers are underweight), but both deletion and duplication carriers show psychiatric features (U-shaped model). The authors may test different models (mirror model, U-shaped model, effect only for deletions, effect only for duplications).

We agree that such models could also yield new discoveries. We tested those models in 21 out of the 26 cohorts that provided such data (including the UK Biobank) and meta-analysed the results. These were already included in Supplementary Table 4, but no reference was made to it indeed. No additional genome-wide significant signal was found, which we added now to the results section (see below).

6-P4: 'For burden analysis all four models were tested'. Please provide more explanations on the models.

We added now a more detailed explanation of the four tested models and refer back to those in the burden analysis:

"As secondary analysis, we also tested U-shaped (assuming the same effect of deletions and duplications at any given locus), deletion-, duplication-only models genome-wide, but these did not yield further significant associations (Supplementary Table 4). All reported CNV effect sizes (unless specified otherwise) represent the impact of one additional copy relative to the population average. For burden analysis all four abovementioned models (mirror, U-shaped, deletion, duplication) were tested."

7-P4: please provide more details about the ref 20 study. The readers may not understand why you test these 59 BMI-associated CNVs.

We added more information on the selection, which can be found in the reference:

"Regions have been defined based on proximity of GWAS SNP^{1,2}, CNVs report⁹, genes from OMIM repository⁸ and from a very recent systematic review of known genes implicated in genetic syndromes with obesity (Table 1 of Kaur et al.¹⁰). Regarding the candidate CNVs/genes or the OMIM regions, all high quality ($r^2>0.5$) probes falling into the regions were selected. For each GWAS SNP we selected all the probes with association results in a +/- 500kb region around the SNP position. The CNV report catalogued 84 BMI and obesity-associated CNVs from research published since 2008 via PubMed search (see Table S2 of the publication⁹). Out of the 84 CNVs we had good quality probes for 48 of them that we subsequently tested. Out of the 79 OMIM regions for weight and BMI, 37 had good quality probes ($r^2>0.5$). The 96 Kaur et al. genes¹⁰ represent 65 regions, out of which 57 are on autosomes and 32 of those contained probes within 10kb with good imputation quality ($r^2>0.5$)."

8-P5: please refer to the recent systematic review on syndromic obesity by Kaur et al. (Obesity Reviews 2017) to

identify syndromic obesity genes. Screening the OMIM repository alone does not provide an exhaustive list of syndromic obesity candidate genes.

We would like to thank the reviewer for pointing us to this very relevant and comprehensive paper. We analysed the set of genes listed in this paper, but found rather poor enrichment. Given the predominantly negative results, we shortened and moved now all candidate analyses to the Discussion upon the request of Reviewer #2 and included the results for the Kaur candidate set there.

9-The synthetic association hypothesis states that partial linkage disequilibrium between a rare variant conferring a strong genetic effect on a complex trait and a common variant may translate into a modest genetic effect of the SNP for the same trait. Considering that several BMI-associated rare CNVs in this study are located in well-established BMI-associated SNP loci (SH2B1, MC4R, FMO5, FLJ25404), it appears essential to exclude the possibility of synthetic association. The authors sometimes provide linkage disequilibrium information between the lead SNPs and CNVs, but it is not sufficient. Even a modest linkage disequilibrium may explain the SNP association. The authors may perform additional analyses (e.g. conditional analyses) to formally demonstrate the independence of rare CNVs and common SNPs associations at these loci.

We have now performed conditional analysis including both the GWAS SNP and the discovered CNV using the UK Biobank data. We found evidence only for the SH2B1-BMI SNP associations to be driven by 16p11.2 CNV probes. We added these analyses to the Results and the following text to the Discussion:

“Our conditional analysis showed that CNV probes in the 16p11.2 region explain a substantial fraction of the association between all previously published SNPs near *SH2B1* and BMI and similarly the association between the SNP near *FLJ25404* and height. None of the remaining associated CNVs showed evidence to tag common SNPs, nor explain known height/BMI-SNP associations. Note, however, that our CNV data are much noisier than SNP calls and thus the measured CNVs are poorer proxies for the true CNV status, which biases the conditional analysis towards the null (no tagging). Still, most of the obtained results are in line with the proposed theory that the majority of the discovered disease-associated common SNPs are not synthetic associations due to rare variant tagging¹¹.”

10-P5: the authors suggest that the association between the 600kb 16p11.2 deletion and BMI was mainly driven by its association with decreased height. It is a bold statement if we consider that 16p11.2 deletion carriers develop a highly penetrant form of monogenic hyperphagic obesity (Walters Nature 2010, Bochukova Nature 2010). Mentioning that the 600kb 16p11.2 deletion was associated with increased BMI and weight and decreased height may be sufficient.

We toned down the statement:

“In addition, we found that while the 220kb deletion increases BMI through increasing weight (by 10.35 kg, $P=5 \times 10^{-9}$), the 600kb deletion does so by both decreasing height (by 5.21 cm, $P=1.1 \times 10^{-14}$) and increasing weight (6.57 kg, $P=5.3 \times 10^{-5}$) (Supplementary Figures 2, 3A-B, 4A-B, 5A-B).”

11-P6: it is not clear if the association results in the UK Biobank are reported separately or are pooled with other GIANT studies.

All association results in the paper refer to the combined sample, unless stated otherwise. We added this to the Introduction:

“All reported results in the paper are based on the full study population, unless stated otherwise (e.g. conditional analysis).”

12-I think that the associations between CNVs and 23-27 phenotypic traits in UK Biobank do not add much to the

study. It is unclear if the authors applied a Bonferroni correction for all the statistical tests performed in UK Biobank, and they do not provide a detailed description of the results. The authors are encouraged to focus on the analysis of the four anthropometric traits in UK Biobank (height, weight, BMI, waist-to-hip ratio), as a majority of other traits (e.g. forced expiratory volume, hand grip strength) are out of context.

We think that these negative pheWAS results are important to show that the identified anthropometric associations are not secondary to some other trait. The aim of this analysis was not to claim new associations with other traits (no Bonferroni correction is necessary), but rather to verify whether these associations are stronger than those with anthropometric traits. This is pointed out in the Discussion:

“These lines of evidence indicate that for most of the discovered CNVs impact anthropometric traits either primarily or in a disease-independent fashion.”

For these reasons we would like to share the results of this analysis with the readers.

13-P6: the cis eQTL analysis is out of context, please remove this section.

We are not sure why the reviewer thinks that this is out of context. Most of those CNVs certainly exert their impact through altering the expression of nearby genes. The fact that we find evidence that some GWAS SNPs have similar effect it can help pinpointing crucial genes and narrowing down the critical region. Our explanation was probably not clear and we try to better explain the reason:

“Genes whose expression is modulated by both trait associated SNPs and CNVs are good gene candidates and can help narrowing down the critical region. To identify such genes, we asked whether known height/BMI-associated SNPs^{1,2} act also as cis eQTLs in blood¹² for the genes located within height/BMI-associated CNVs.”

14-P7: please double-check that all the references are in superscript, and that the font size is unique. When you refer to the SNP near MC4R associated with BMI, please cite Loos et al., Nat Genet 2012.

We double-checked the references and added Loos et al 2008 on MC4R and Loos 2011.

15-P7: it is surprising that the CNV near MC4R is not associated with height (rare mutations and SNPs in/near MC4R are, see Loos et al., Nat Genet 2012). This deserves more discussion.

We have carefully checked the association with height, but found only a very weak association (provided in Supplementary Table 4): Each additional copy of the region reduces height by 3.16cm ($P=1.41 \times 10^{-2}$). We added the following to the results section:

“While rare height-increasing *MC4R* variants¹³ have been previously reported, we found no height-effect of any CNV probes in this regions.”

16-The authors mention that a majority of SNPs associated with anthropometric traits are also associated with schizophrenia in literature. This deserves more discussion.

There may be a misunderstanding: we only pointed out that surprisingly many among our BMI-associated CNVs (1q21.13q29, 7q11.23, 16p11.2) also associates with schizophrenia. We do not claim anything about SNPs, neither claim a general trend in the literature.

17-P8: please remove the section 'Associations with suggestive significance'. These associations do not pass the multiple testing adjusted association threshold and are likely to be false-positive.

We removed the description of these regions from the paper and only extended one sentence of the Discussion to mention them, in case researchers are interested:

"Furthermore, we identified 7 CNVs significantly associated with at least one trait and three additional CNV regions have a close to genome-wide significant effect on one of the four traits (Supplementary Table 25)."

18-P9: the lack of replication of previous CNV associations can also signify that the associations reported in literature are spurious (please cite Walters et al., PLOS One 2013). Please update the text accordingly.

Indeed, this can also be an important reason. We added the Walters paper.

19 -P9: 'We believe that instead of collecting more samples from the UK and Finland, collecting larger population-based cohorts from other countries could be a more efficient way to discover novel trait-associated CNVs with larger effects'. Please update this over simplistic statement. First, these actions are not mutually exclusive. Second, genotyping genome-wide DNA arrays is more challenging and expensive than collecting biological samples in large populations. Third, the authors may also mention to perform CNV association studies in other ethnic groups.

We improved the sentence according to the reviewer's suggestion:

"Therefore, we believe that, in the future, collecting larger, genotyped population-based cohorts from other countries and ethnicities could be an efficient way to discover novel trait-associated CNVs with larger effects."

20-P9: 'None of the associated CNVs seems to tag common SNPs, nor explain known height/BMI associations'. This is an overstatement. The data presented in the Results section do not enable the authors to make any conclusion.

Correct, even conditional analysis cannot give a definitive answer. We added the following statement:

"Our conditional analysis showed that none of the associated CNVs seems to tag common SNPs, nor explain known height/BMI SNP associations. However, it is very likely that our CNV data is still quite noisy and the measured CNVs are just proxies for the true CNV status, which biases the conditional analysis towards the null (no tagging). Still, the obtained results are in line with the proposed theory that the majority of the discovered disease-associated common SNPs are not synthetic associations due to rare variant tagging¹¹."

21-P9: please replace 'an atlas of CNV map in the general population' by 'an atlas of CNV map in the general European ancestry population'.

We changed it to "an atlas of CNV map based on a large general population of European ancestry".

Reviewer #2 (Remarks to the Author):

Summary

CNVs on anthro traits in ~191k individuals from 26 cohorts identified 5 novel associations and 2 known ones. The discovered associated CNVs are rare, and have large effect sizes. They used PENNCNV on Illumina or Affy Axiom to generate CNV calls and defined the threshold of significance to be 1.7×10^{-6} .

Major concerns

1. I wonder about the utility of doing a genome-wide association study of CNVs but using the more liberal study-wide threshold for significance of 1.7×10^{-6} . If the CNV is tagging a single nucleotide variant, it would not be described in a GWAS but would be here.

Indeed, this significance level is higher than we are used to for SNP analysis. It corresponds to ~30,000 independent tests. The reason for this is that CNV probes show much less variability than SNPs do. First, vast majority of the genome (>75%) in our study of >191,000 individuals is copy neutral in at least 99.99% of the samples. We only analysed probes with deletion and duplication combined frequency >0.01%. Second, the correlation between close-by probes in terms of copy numbers is far larger than it is for SNPs. Almost all of the reliably detectable CNVs ($Q_S > 0.7$) are >10kb with the mode >100Kb, containing ~100 probes on average. This would mean as if SNPs within 100kb were in perfect LD, which is clearly not the case, most SNP LD dies of by 100kb distance. More precisely, we calculated the effective number of independent tests for the 242,022 probes that showed non-copy neutral state in more than 1 in 10,000 samples. For this first we used the Hypergenes cohort ($n=2,930$ samples) and the procedure proposed by Gao *et al.* 2008 (see Online Methods), which exploits the local LD structure between probe copy number states. We found that the ratio between total number of (copy-number variant) probes and the corresponding number of equivalent independent probes [denoted by f in the paper] is 8.23. This projected the 242K probes in the full data set to be equivalent to 29,412 independent tests (see Methods) giving rise to a Bonferroni corrected genome-wide threshold of 1.7×10^{-6} . To make sure that our calculations are robust, we have now repeated the experiment using the UK Biobank samples. For each batch we calculated the ratio across the 22 autosomes. The median ratio across the 33 batches was 13.83 ($CI_{95\%}=[8.20, 20.17]$). This means that our initial estimate was even slightly conservative and the genome-wide significant threshold may be increased to 2.85×10^{-6} . We still kept the old threshold to remain more conservative. Moreover, the most recent genome-wide CNV association paper⁸ used a much milder $\sim 1.7 \times 10^{-4}$ threshold for genome-wide significance (at 5% FWER).

We updated the Methods accordingly:

"This indicated the strength of dependence between CNV probes, $f = N_{eff}/N_{non-zero}$. We obtained a ratio of $f=8.23$ and applied this scaling constant to the 242,022 probes tested in our meta-analysis study, yielding 29,412 independent tests and subsequently 1.7×10^{-6} genome-wide significant threshold. To ensure robustness, we repeated the same analysis for each of the 33 batches of the UK Biobank samples and obtained a slightly less stringent ratio (median $f=13.83$, $CI_{95\%} = [8.20, 20.17]$), but we preferred to use the more conservative threshold of 1.7×10^{-6} ."

2. Definitely delete the section on "Associations with suggestive significance"

We removed it and only refer to the Supplementary Table in the Discussion in case readers are interested to go beyond the main findings.

3. How can you really tell between population-specific CNVs and differences in CNV calling due to array type? particularly for the UK biobank? This explanation at the end of the results section is not convincing.

Good point. We have now checked the frequency of this CNV in the UK Biobank (0.028% (del), 0.005% (dup)) as well as in other UK cohorts genotyped on Illumina arrays (0.018% (del), 0.009% (dup)) and noted that both were consistently higher, than non-UK samples (0.006% (del), 0.005% (dup)). Still, of course, the difference maybe partially array specific. We have admitted this weakness:

“Note that the frequency of the *MC4R* CNV both in the UK Biobank (0.028% (del), 0.005% (dup)) and in other UK cohorts genotyped on Illumina arrays (0.018% (del), 0.009% (dup)) is consistently higher, than the frequency in non-UK samples (0.006% (del), 0.005% (dup)). Thus, the observed frequency difference is at least in part not due to array-effect.”

4. There is no discussion of which region(s) in the deletions might be functional. Any genes? Any critical regulatory elements?

Since the discovered CNVs in every case fully include multiple genes (all listed in Table 1) and their regulatory regions, we do not believe that any meaningful statement could be made on the particular functional impact of these CNVs. We mentioned in the Discussion that in most cases we already struggle to pinpoint the critical gene. As opposed to SNP-based GWAS where this approach would indeed be necessary, here when entire genes are deleted the consequence in terms of function is clearer, but the key genes are difficult to find.

5. The deletion burden analysis needs more detail.

We agree, this was missing, now added to the methods:

“For each sample we calculated the total number of (imputed) CNV probes showing deviation from the copy neutral state. To account for uncertainty in the calls and to avoid arbitrary thresholding, we used the absolute QS of each probe (for the U-shaped model) and summed them up for the whole genome. For the other models (deletion-, duplication-only) we used the respective modifications (minus deletion QS, duplication QS). We then ran a linear regression between the total burden score and the various traits and meta-analysed the results from the 26 cohorts.”

6. The QQ plots look inflated - what were the lambdas for the CNV association tests?

Those are the QQ-plots for the candidate regions only. For clarity, we now report the QQ-plots for the whole genome (Supplementary Figure 22). As it is the case for well-powered GWAS studies on highly heritable traits, the genomic lambda value reflects mostly polygenicity. To prove this we adapted the LD score regression¹⁴ to the copy-number values of the probes in the UK biobank. We added the corresponding text to the Discussion:

“Overall the genome-wide P-values showed good adherence to the null distribution (Supplementary Figure 22). For well-powered GWAS studies on heritable traits (e.g. height², menarche¹⁵) high genomic control lambda value rather reflects true polygenic signal than uncorrected population stratification¹⁴. This was the case for our study too: while we observed inflated genomic lambda coefficients ($\lambda = 1.16$ (height), $\lambda = 1.12$ (weight), $\lambda = 1.08$ (BMI) and $\lambda = 1.05$ (WHR)), upon applying LD score regression in the UK Biobank sample the intercept terms revealed no unaccounted population stratification ($\lambda_{LD} = 0.971$ (height), $\lambda_{LD} = 1.005$ (weight), $\lambda_{LD} = 0.993$ (BMI), $\lambda_{LD} = 0.942$ (WHR)). “

7. I wonder if so many non-significant association results in Table 1 is useful. Why not make this a supplemental table and just give the genome-wide (or study-wide) significant results only with trait listed?

The Table only contains genome-wide significant associations. Of course we could reduce the size of the table, but that would make it less clear. If space constraints occur from the journal, we are happy to reduce this table to the bare essentials as suggested by the reviewer.

8. I would delete the section on candidate CNVs - not significant.

We agree, these results are not particularly convincing, thus moved them to the Supplement (Figure S1).

Minor concerns

9. "which share genetic basis with developmental- and psychiatric disorders." not sure if this implies the genes are in common between anthro and psych disorders or the genetic architecture is similar. **Yes, only the loci, not necessarily the genes are shared. We changed the word "basis" to "loci", which we clearly demonstrated.**

10. "We did not observe any CNV burden effect on the weight" what weight? Weight the trait?

Yes, we changed the ambiguous sentence to:

"We did not observe any CNV burden effect on human weight."

11. 59 BMI-associated CNVs manuscript 20? Are these previously reported at genome-wide significance?

Not all, some are candidate studies. We better describe now that study and specifically refer to their Supplementary Table 2, where the details are given. We also include now a newer and more complete literature review by Kaur et al. 2016.

12. Grammatical errors "Further three CNVs" and "Burden of rare CNV" "The first, 220kb BP2-BP3 deletion" etc.

We corrected these and had a careful check thorough the manuscript.

13. "And finally one deletion, encompassing 204 BBOX1 and FIBIN genes, seems to be particularly frequent in the Finnish population (0.89% vs 0.02% in 205 the non-Finnish cohorts)." Is it associated with anything?

We described it later in the Results, but added that a rare coding variant in FIBIN is known to lower height.

14. We could not narrow down the BMI association signal to the previously proposed SH2B1 (lowest $P=7.7 \times 10^{-8}$) as it spreads out to other genes, including SPNS1 and LAT (lowest $P=5.3 \times 10^{-8}$ 230) (Figure 3) - please revise "as it spreads out"??

Done, we use the word "cover" instead of "spread out to".

Reviewer #3 (Remarks to the Author):

The manuscript by Mace, Kutalik and colleagues entitled “CNV-association meta-analysis in 191,161 adults reveals new loci associated with anthropometric traits” reports the largest ever study of copy number variants (CNVs) with a complex trait. Taking advantage of height, weight and waste-hip ratio measurements from large cohort studies the authors have carried out a systematic analysis of the effects of CNVs on these anthropometric traits. Given that most CNVs that are known to influence height and weight are quite rare in the general population, it would be an accomplishment simply to identify a few credible associations. The sample size of this study, however, provided sufficient power to demonstrate a significant influence of CNV burden on height and weight and to identify seven loci that survive genome-wide multiple testing correction. These rare variants have large effects on height and weight (and neurodevelopment), providing further evidence that physical and cognitive traits have a shared genetic basis. Rare variants were identified at 3 loci that have been identified in previous GWAS (MC4R, FIBIN, FMO5), suggesting that common and rare variants of these genes influence these traits.

The data processing and statistical methods are highly rigorous and incorporate a number of clever innovations. The evidence supporting the association of CNVs with anthropometric traits is strong.

Thanks for the positive comment.

I have several minor concerns to be addressed.

1. Data availability. After logging in to the website (<https://molgenis59.gcc.rug.nl>), the browser displays only genes, repeats and conservation tracks. The track labeled “CNVs” is empty. This needs to be fixed. Will a bulk download of the CNV calls also be available?

Sorry the webpage in the meantime has been moved to <https://cnvcatalogue.bbMRI.nl> This version shows the CNV track. Supplementary Table 26 has the full bulk data, as indicated in the main text. However, we have now added association effect sizes and P-value for the 4 traits to the Supplementary Table 26.

2. Line 107: There are few examples of robust associations between rare copy number variants and complex traits. Do you mean “anthropometric traits”. In addition to many chromosomal rearrangements that are associated with Mendelian disease, there are dozens that are associated neuro-psychiatric disorders (which qualify as “complex”).

Indeed, we were not specific enough. We meant complex continuous human traits and we do not include diseases. Or if we were, we would focus on common diseases.

3. CNV calling: methods do not state whether any filters were applied on quality score, CNV length or number of probes. Such post-processing steps are typically done. If no such steps were taken, this should be clarified.

We have now clarified it in the Methods:

"Since the QS accounts for various CNV characteristics (length, number of probes, etc.) we did not apply any filtering on these scores, which was shown to be the most powerful strategy for association¹⁶. However, probes with low imputation quality (see below) are filtered out in each cohort. "

4. If quality control was handled exclusively within the statistical model, by using the quality score as a genotype value, this represents an innovation that departs somewhat from the norm. Some additional attention to this would be helpful. For instance, If there are many low-quality calls overlapping a probe, the MACH r-squared value will approach zero and that probe will be filtered out. Correct? Assuming, the probe passes this filter, then how is the dosage of individuals in the cohort (Gc) calculated? How is the quality score used to calculate a weighted genotype value?

Yes, indeed, the approach represents an innovation, which has been described in our earlier publication [Mace *et al.* Bioinformatics 2016, ref16]. This is also described in the CNV calling and Association subsections of the Methods. As correctly pointed out by the reviewer, if a probe is covered by low quality CNV calls the r-squared will approach zero. The QS is the same as the relative dosage compared to the copy-neutral state. E.g. QS = -0.5 means that it is 50% a (single deletion), i.e. its expected copy number dosage is 2 with 50% probability and 1 with 50% probability, thus the expected dosage is 1.5, which is -0.5 dosage lower than the copy-neutral (2 copy) state. We added this explanation to the Methods/CNV calling.

5. The basic statistical meta-analysis approach makes sense, but I am curious how this approach to filtering and meta-analysis influence the association results. As the allele frequencies become very high (MAF >5%), are these probes more likely to be filtered out? Or are common CNVs well represented?

We apply only one filtering step: for a given cohort we exclude a probe if it has low (<0.1) r²-hat value. The filtering does not act differently on CNVs with different frequency. As we showed in our previously published paper [ref16], more common CNVs tend to have very slightly higher r²-hat and hence have higher chance to be kept. Thus, we believe both common and rare CNVs are treated equally.

6. Related to the previous question: One of the more puzzling results of this study is the LACK of association of common CNVs. The logical conclusion of this is that the effects are extremely modest. This is surprising though considering that common SNP associations are readily detectable in the same sample. Are common CNVs well ascertained? Well represented (see previous comment)? Perhaps there are just so few common alleles detectable with genotyping arrays?

Yes, these are very interesting questions. As opposed to the MAF distribution of SNPs, the MAF of CNVs is much more centred on zero. As from Supplementary Table 26 can be calculated, among the 250K copy number variant probes (i.e. MAF>0.01%) the maximum frequency was 12.7% and the median frequency is 0.014% and less than 8% of the probes have frequency >1:1000. Common CNVs are very rarely detected, which may be a consequence of the technology or reflecting true biology. We added this important follow-up to the Discussion:

"This may be explained by the massive shift of CNV frequency spectrum compared to that of SNVs: based on CNV calls from >191,000 samples we observed that more than 92.4% of the CNVs are present in less than 1 in 1,000 samples and 99.4% of them are rare (<1%). We are unsure whether the reason for the very low number of common CNVs is due to the detection technology or reflecting the nature of the underlying genomic event."

7. Line 393: We believe that False CNV calls do not translate into false positive findings. Probably not, that's the

beauty of a “case-only” design. Issues of poor data quality may affect specific dataset, but it would be unlikely for false CNV calls in the sample to correlate with clinical measures. In this kind of cohort study, CNV calling error would not lead to false-positive associations BUT it could diminish the evidence for association. This is something that should be considered when interpreting some of the negative results (lack of association of common CNVs or lack of replication of previous CNV findings.)

We acknowledge it now in the Discussion:

“Thus, we believe that false CNV calls do not translate to false positive findings, but of course can substantially reduce statistical power.”

We also added that one reason for non-replication of previous findings may be due to low power.

8. Supplementary Figures: Frequency units are %?

Yes, in all Tables. We specifically indicate it on every instance.

References

1. Locke, A.E. *et al.* Genetic studies of body mass index yield new insights for obesity biology. *Nature* **518**, 197-206 (2015).
2. Wood, A.R. *et al.* Defining the role of common variation in the genomic and biological architecture of adult human height. *Nat Genet* **46**, 1173-86 (2014).
3. Marouli, E. *et al.* Rare and low-frequency coding variants alter human adult height. *Nature* (2017).
4. Yang, J. *et al.* Genetic variance estimation with imputed variants finds negligible missing heritability for human height and body mass index. *Nat Genet* **47**, 1114-20 (2015).
5. Gamazon, E.R., Cox, N.J. & Davis, L.K. Structural architecture of SNP effects on complex traits. *Am J Hum Genet* **95**, 477-89 (2014).
6. Jacquemont, S. *et al.* Mirror extreme BMI phenotypes associated with gene dosage at the chromosome 16p11.2 locus. *Nature* **478**, 97-9102 (2011).
7. Zufferey, F. *et al.* A 600 kb deletion syndrome at 16p11.2 leads to energy imbalance and neuropsychiatric disorders. *J Med Genet* **49**, 660-8 (2012).
8. Contribution of copy number variants to schizophrenia from a genome-wide study of 41,321 subjects. *Nat Genet* (2016).
9. Peterson, R.E. *et al.* On the association of common and rare genetic variation influencing body mass index: a combined SNP and CNV analysis. *BMC Genomics* **15**, 368 (2014).
10. Kaur, Y., de Souza, R.J., Gibson, W.T. & Meyre, D. A systematic review of genetic syndromes with obesity. *Obes Rev* (2017).
11. Wray, N.R., Purcell, S.M. & Visscher, P.M. Synthetic associations created by rare variants do not explain most GWAS results. *PLoS Biol* **9**, e1000579 (2011).
12. Westra, H.J. *et al.* Systematic identification of trans eQTLs as putative drivers of known disease associations. *Nat Genet* **45**, 1238-43 (2013).
13. Loos, R.J. The genetic epidemiology of melanocortin 4 receptor variants. *Eur J Pharmacol* **660**, 156-64 (2011).
14. Bulik-Sullivan, B.K. *et al.* LD Score regression distinguishes confounding from polygenicity in genome-wide association studies. *Nat Genet* **47**, 291-5 (2015).
15. Day, F.R. *et al.* Genomic analyses identify hundreds of variants associated with age at menarche and support a role for puberty timing in cancer risk. *Nat Genet* (2017).
16. Mace, A. *et al.* New quality measure for SNP array based CNV detection. *Bioinformatics* (2016).

REVIEWERS' COMMENTS:

Reviewer #1 (Remarks to the Author):

The authors addressed all my comments, thank you.

Reviewer #2 (Remarks to the Author):

I am satisfied with the comments from the authors, with the exception of Table 1. I strongly feel that inclusion of many non-significant pvalues in Table 1 takes away from the message and interpretation of the results.

However, I could not review Table 1 again as it does not appear to be available in the documents for review.

Reviewer #3 (Remarks to the Author):

The authors have adequately addressed my concerns.